# Phosphosite Scanning reveals a complex phosphorylation code underlying CDK-dependent activation of Hcm1

Michelle M. Conti [1], Rui Li [1], Michelle A. Narváez Ramos [1], Lihua Julie Zhu [1,2], Thomas G. Fazzio[1] & Jennifer A. Benanti [1] ✉

Ordered cell cycle progression is coordinated by cyclin dependent kinases (CDKs). CDKs often phosphorylate substrates at multiple sites clustered within disordered regions. However, for most substrates, it is not known which phosphosites are functionally important. We developed a high-throughput approach, Phosphosite Scanning, that tests the importance of each phosphosite within a multisite phosphorylated domain. We show that Phosphosite Scanning identifies multiple combinations of phosphosites that can regulate protein function and reveals specific phosphorylations that are required for phosphorylation at additional sites within a domain. We applied this approach to the yeast transcription factor Hcm1, a conserved regulator of mitotic genes that is critical for accurate chromosome segregation. Phosphosite Scanning revealed a complex CDK-regulatory circuit that mediates Cks1-dependent phosphorylation of key activating sites in vivo. These results illuminate the mechanism of Hcm1 activation by CDK and establish Phosphosite Scanning as a powerful tool for decoding multisite phosphorylated domains.

Progression through the cell division cycle requires the coordinated function of hundreds of proteins. This coordination is achieved in part through the phosphorylation of numerous substrates by cyclin-dependent kinase (CDK) complexes[1–5]. This critical mechanism of cell cycle control is conserved from yeast to humans[6,7]. Phosphorylation of CDK substrates can trigger a variety of actions, including substrate degradation or activation, interactions of substrates with other proteins, and re-localization of substrates to different sub-cellular compartments. These changes have widespread implications in cell cycle control, including the regulation of key phase transitions and the expression of cyclically transcribed genes.

A common feature among CDK substrates is that they are often phosphorylated on multiple sites. CDKs are serine and threonine-directed kinases that efficiently phosphorylate an optimal consensus motif (S/T-P-x-K/R) in vitro[8–10]. However, a minimal consensus motif (S/T-P) is frequently phosphorylated in vivo. These consensus motifs often occur in clusters and are enriched in intrinsically disordered regions (IDRs)[3,11,12]. Despite these common features, the numbers and/or positions of phosphates required to elicit functional changes vary considerably among CDK substrates. Some substrates contain many CDK phosphorylation sites, but functional changes depend on the phosphorylation of only one or a few specific sites. Phosphorylation at a specific site can be required to trigger structural changes that impact protein function[13–15], or to generate a linear binding epitope[16,17]. Conversely, in a different group of substrates, phosphorylation at many sites is required to reduce the overall charge of a domain and elicit functional changes[18]. For most CDK substrates, the CDK-regulatory circuitry remains poorly understood, including (1) the fraction of phosphosites required to regulate protein function and (2) which particular sites, among many that are phosphorylated by CDKs, have regulatory function.

Determining how individual phosphosites contribute to protein regulation has remained a significant challenge because it requires generating large numbers of mutant proteins for any given substrate

[1]Department of Molecular, Cell and Cancer Biology, University of Massachusetts Chan Medical School, Worcester, MA 01605, USA. [2]Program in Bioinformatics and Integrative Biology, University of Massachusetts Chan Medical School, Worcester, MA 01605, USA. ✉e-mail: jennifer.benanti@umassmed.edu

and evaluating the effects of each mutant one by one. Although it may be feasible to generate a series of mutants in which each phosphosite is mutated individually, this strategy only examines the importance of each phosphosite in an otherwise wild-type context, potentially missing effects observed when combinations of phosphosites are mutated. One streamlined approach that has been developed to systematically examine the effects of mutations on protein function in a pooled format is deep mutational scanning[19–22]. Most deep mutational scanning approaches analyze all possible amino acid changes at each residue within a domain, which provides high-resolution information about protein structure. Here, we modified deep mutational scanning to focus specifically on individual phosphosites within a multiphosphorylated domain, and to determine how phosphorylation at each site impacts protein function in vivo. Our modified approach, called Phosphosite Scanning, uses bulk competition assays to measure the cellular fitness outcomes caused by the mutation of phosphorylation sites to unphosphorylatable or phosphomimetic residues, in all possible combinations. By comparing the fitness effects of proteins with different combinations of wild-type (WT), unphosphorylatable and phosphomimetic residues, it is possible to determine which phosphosites contribute to protein regulation, the combinations of phosphosites that are necessary and sufficient for normal activity, and how phosphorylation at a particular site impacts phosphorylation at additional sites within a domain.

To examine how CDK can regulate protein function through multisite phosphorylation, we applied Phosphosite Scanning to the yeast forkhead family transcription factor Hcm1. Like its human homolog, FoxM1, Hcm1 activates expression of genes that are required for mitotic spindle function and accurate chromosome segregation[23,24]. Notably, mutations in Hcm1 impact cellular fitness. Mutation of all CDK phosphoacceptor sites in Hcm1 to alanine inactivates the protein and reduces cellular fitness[25]. Conversely, mutation of all eight phosphoacceptor sites within the transactivation domain (TAD) to two glutamic acid residues—mimicking the charge of a phosphate—results in elevated activity and increased fitness compared to WT[26]. These phosphorylation-dependent fitness effects make the Hcm1 TAD an ideal model to investigate the mechanism of phosphoregulation by CDK.

Phosphosite Scanning of Hcm1 revealed several features of activation that highlight the complex regulation of multisite phosphorylated domains. Remarkably, we found that non-overlapping combinations of phosphomimetic mutations were able to activate Hcm1. Phosphomimetic mutations at two central phosphosites, T460 and S471, had the greatest impact on Hcm1 activity and, on their own, could restore WT activity when all other phosphosites were changed to alanine. However, phosphomimetic mutations at all sites except T460 and S471 led to similar activation, indicating a mechanism in which both the position and the number of CDK sites are important for activation. We also found that phosphorylation of three N-terminal threonine residues within the TAD (T428, T440, and T447) was necessary for the phosphorylation of T460 and S471. Our results suggest that CDK sites within the TAD are efficiently phosphorylated via the CDK phosphoadaptor subunit Cks1, which interacts specifically with phosphothreonines to promote increased phosphorylation. In sum, our results reveal the mechanism of Hcm1 activation by CDK and establish Phosphosite Scanning as a powerful approach that can be used to interrogate multisite phosphorylated domains.

## Results

### Phosphorylation of the Hcm1 TAD regulates fitness
Hcm1 is a Cln2/Cdk1 substrate[27] that is highly phosphorylated as cells enter S phase (Supplementary Fig. 1a, b). Hcm1 regulation by CDK is complex: it contains 15 minimal CDK consensus sites that regulate both its degradation and transcriptional activity[25]. Like other CDK substrates, most of these sites fall within IDRs (Fig. 1a). Of the 15 consensus

sites, a cluster of three sites in the N-terminus constitutes a phosphodegron that, when phosphorylated, signals for its degradation by an SCF-family ubiquitin ligase[25]. A second cluster of eight sites is located within the TAD and is required for Hcm1 to activate the transcription of target genes. However, the specific requirements for phosphoregulation of the TAD are not understood.

Previously, mutation of all 15 CDK phosphoacceptor sites in Hcm1 to alanine was shown to reduce cellular fitness in a pairwise, co-culture assay[25]. To investigate the effects of blocking phosphorylation exclusively within the TAD, we mutated the eight sites in this region to alanine (hcm1-8A) and assayed the fitness of this strain compared to wild type (WT) in a co-culture assay. Low copy centromeric plasmids expressing WT HCM1 or hcm1-8A from its endogenous promoter were transformed into strains expressing either mutant (non-fluorescent) or WT GFP, respectively, and in which the galactose-inducible GAL1 promoter was integrated upstream of the genomic copy of HCM1. At the start of the assay, equivalent amounts of each strain were diluted into dextrose-containing medium to shut off the expression of endogenous HCM1 (Fig. 1b). Co-cultures were then sampled and diluted every 12 h for a total of 72 h to prevent cells from reaching saturation and exiting the cell cycle. At the end of the experiment, the percentage of each strain in the population at each timepoint was quantified by flow cytometry. As a control, co-culture assays with WT strains confirmed equal fitness of strains expressing functional or mutant GFP markers (Fig. 1c). Consistent with a role for TAD phosphorylation in Hcm1 function, hcm1-8A cells were dramatically less fit than WT cells (Fig. 1d).

We next examined an Hcm1 phosphomimetic mutant in the same assay. Replacement of all S/T-P motifs in the TAD with two glutamic acid residues (E-E, hcm1-8E) was previously shown to result in a gain of function allele that exhibited increased recruitment to target gene promoters and elevated expression of target genes[25]. This effect was observed when all eight S/T-Ps were replaced with E-Es, but not when S/T-Ps were mutated to E-Ps, suggesting that it is the charge of the phosphate that is important for Hcm1 activation. Interestingly, the phosphomimetic mutant, hcm1-8E, showed the opposite phenotype as the hcm1-8A mutant and conferred a fitness benefit relative to WT, similar to previous results (hcm1-8E was previously referred to as hcm1-16E[26]) (Fig. 1e). We sought to leverage these opposing effects on fitness to investigate the contribution of each CDK phosphorylation site within the TAD to Hcm1 function.

### Phosphosite Scanning: decoding Hcm1 phosphoregulation through pooled fitness assays
We developed an approach, called Phosphosite Scanning, in which bulk fitness screens are used to interrogate all possible combinations of phosphorylated sites within a multiphosphorylated domain. In contrast to saturating mutagenesis approaches[28,19] this method focuses on amino acid substitutions that mimic or prevent phosphorylation. The fitness of strains expressing phosphomutant proteins can be measured by monitoring the frequency of all mutants in the population over time using deep sequencing. By using fitness as a proxy for protein activity, the impact of phosphorylation at each site on Hcm1 activity can be quantified. Screens utilizing only unphosphorylatable and phosphomimetic mutations quantify the activity of proteins with a fixed phosphorylation state. In addition, the inclusion of WT sites along with phosphosite mutations allows us to infer which sites are phosphorylated in vivo (described below).

We first designed a set of phosphomutant alleles such that each S/T-P site was mutated to either unphosphorylatable A-P or phosphomimetic E-E, in all possible combinations, to generate a pooled library of 256 distinct phosphomutants (referred to as the A/E library, Fig. 2a). Gene fragments containing phosphosite mutations were assembled by annealing and extending a mixture of overlapping mutagenic oligonucleotides that cover all eight CDK sites within the TAD (Supplementary

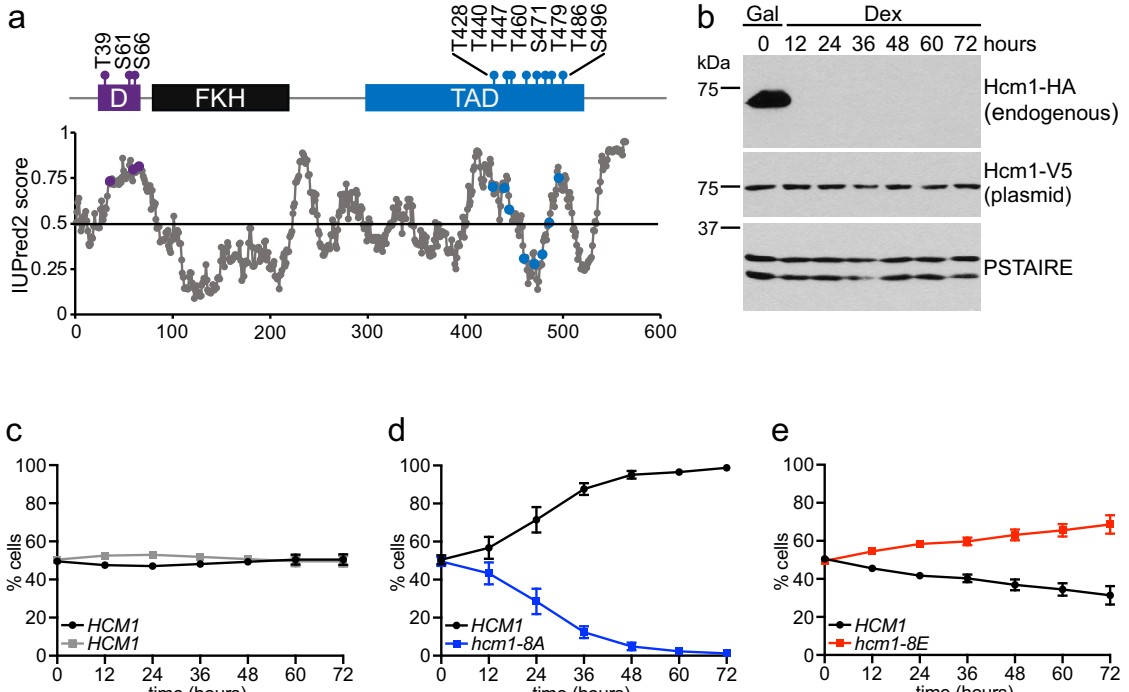

**Fig. 1 | Phosphorylation of the Hcm1 TAD regulates fitness. a** Schematic showing previously characterized CDK consensus sites (S/T-P), domains and predicted disordered regions. D phosphodegron, FKH forkhead domain, TAD transactivation domain. IUPred2 disorder prediction shown with phosphorylation sites in purple (D) and blue (TAD). **b** Western blot showing expression of the indicated proteins at the timepoints shown. Strains were initially grown in galactose (Gal) to allow expression of the 3HA-tagged WT protein from the genome and were shifted to dextrose (Dex) to repress WT expression at the start of the experiment. Plasmid-expressed Hcm1 proteins were detected with an antibody that recognizes a C-terminal 3V5 tag, PSTAIRE is shown as a loading control. Representative images from $n = 3$ biological replicates are shown. **c**–**e** Strains expressing the indicated *HCM1* alleles from plasmids were co-cultured and the percentage of cells expressing each allele was determined at the timepoints indicated. In all, 5000 cells from each timepoint were analyzed. Black and gray represent cells expressing wild-type *HCM1* (**c**–**e**), blue represents cells expressing *hcm1-8A* (**d**), and red represents cells expressing *hcm1-8E* (**e**). Shown is an average of $n = 3$ experiments, error bars represent standard deviations.

Fig. 2a). The fragments were then ligated and cloned into a low copy expression vector containing the *HCM1* promoter. Despite variation in codon usage, phosphosite mutations did not affect Hcm1 protein levels (Supplementary Fig. 2b). The A/E library, which included a construct expressing the WT gene, was then transformed into a *GAL1p-HCM1* strain in galactose-containing medium, similar to the pairwise co-culture assay described above (Fig. 2b). This ensured that WT Hcm1 was present in cells prior to the start of the screen to prevent selection against strains expressing non-functional or lowly functional *HCM1* mutants. The population was sampled at the start of the screen and cells were then shifted to dextrose-containing medium to shut off endogenous *HCM1* expression. The culture was sampled and diluted every 12 h for a total of 72 h to prevent the cells from reaching saturation and exiting the cell cycle. The abundance of each Hcm1 mutant in the population at each timepoint was then determined by Illumina sequencing. The screen was performed in triplicate, and replicates were highly correlated (Supplementary Fig. 3a).

To evaluate changes in the abundance of individual mutants, we calculated the log2 fold change in the fraction of reads, with respect to the initial population, for each mutant over time. Changes were then normalized to WT (Fig. 2c). Importantly, the *hcm1-8A* mutant was depleted over time while the *hcm1-8E* mutant was enriched, validating our pooled approach. Clustering of the mutants based on the number of phosphomimetic mutations revealed several important features of phosphoregulation (Fig. 2c). First, fitness tended to increase with the number of phosphomimetic substitutions. However, when we examined a cluster of mutants that contained the same number of phosphomimetic sites (such as 4E, 4A) a wide range of fitness values was observed. This data rules out the possibility that a minimum number of

phosphates is all that is needed for activation and argues that phosphorylation at particular sites can have a greater impact on activity and fitness. Importantly, when we considered the group of mutants that have only one phosphomimetic mutation (1E, 7A), we saw that no single site was sufficient to confer WT fitness or better, suggesting that no single site is sufficient for activity. Conversely, all mutants that have only one unphosphorylatable mutation exhibited high fitness (7E, 1A), demonstrating that no single site is absolutely required for fitness (Fig. 2c and Supplementary Fig. 2c, d).

We next calculated a selection coefficient, or fitness score, for each mutant. The selection coefficient is the slope of the best-fit line formed when the normalized log2 fraction of reads versus time is plotted for each mutant[29]. All values are then normalized to WT so that mutants that are less fit than WT, like *hcm1-8A*, have a negative selection coefficient, whereas mutants that are more fit than WT, like *hcm1-8E*, have a positive selection coefficient (Fig. 2d). Mutants with fitness like WT have selection coefficients close to zero. Notably, selection coefficients determined from our screen were comparable to those calculated from pairwise competition assays (Supplementary Fig. 2e). To quantitatively examine how the number of phosphomimetic substitutions contributes to fitness, we calculated median selection coefficients for all mutants with a given number of phosphomimetic substitutions (Fig. 2e). These results confirmed that fitness increased with the number of phosphomimetic substitutions, with almost all mutants with six or more sites displaying increased fitness compared to WT.

Next, we investigated the impact of phosphorylation at individual sites by calculating the median of the selection coefficients of all mutants with a specific substitution at each site (Fig. 2f). Overall,

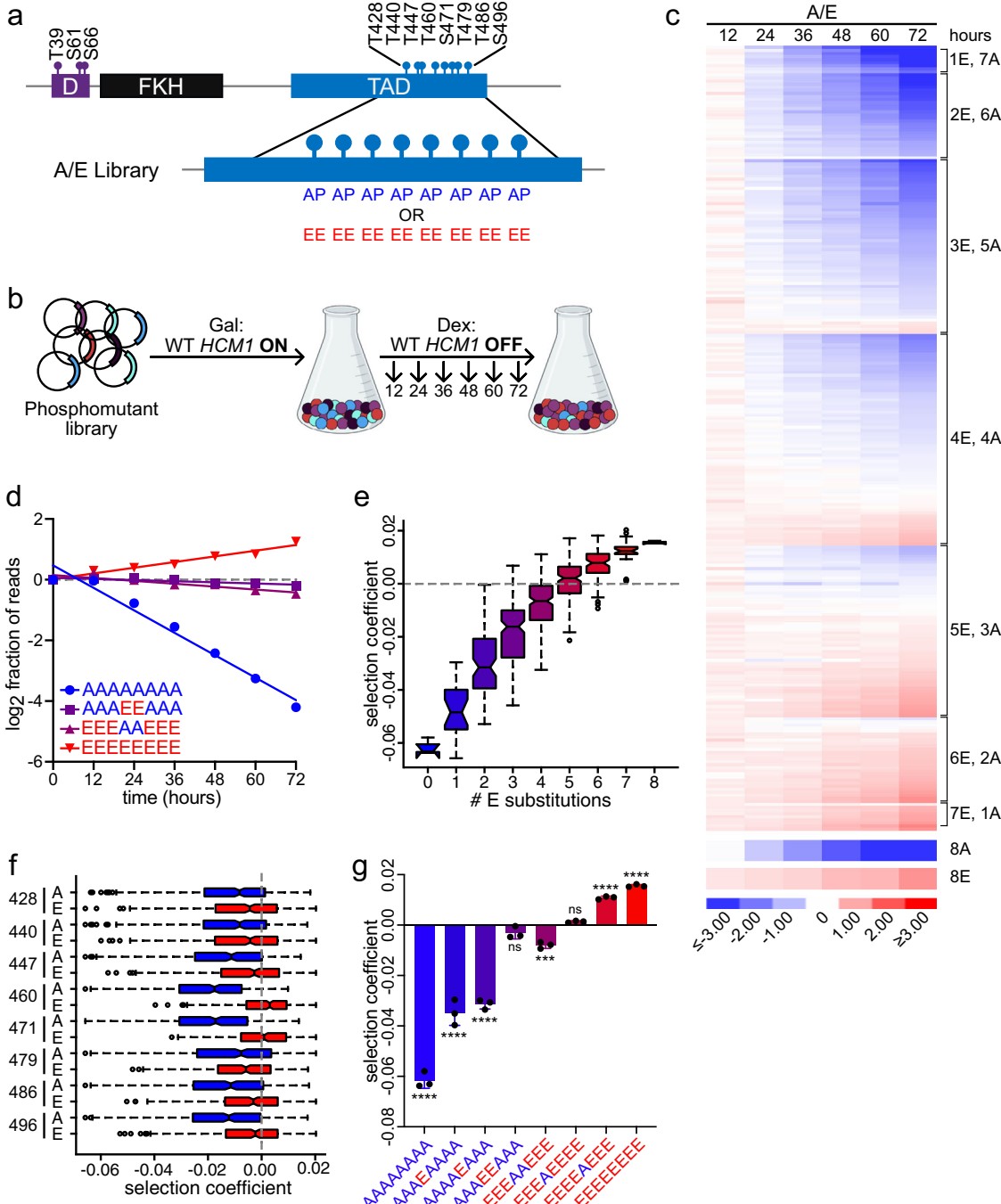

**Fig. 2 | Decoding Hcm1 phosphoregulation with Phosphosite Scanning.**
**a** Schematic of phosphomutant library design. Each S/T-P site within the TAD was mutated to either unphosphorylatable A-P or phosphomimetic E-E in all 256 possible combinations. Created with Biorender.com. **b** Schematic of the Phosphosite Scanning approach, see text for details. **c** Results from the A/E Phosphosite Scanning screen. Each row represents a mutant, shown is the log2 fold change in normalized read counts with respect to time zero for each mutant, all mutants have been normalized to WT. Blue indicates depletion, red indicates enrichment. Shown is an average of $n = 3$ biological replicates. See Supplementary Figs. 2b–d and 3a for additional analyses and correlation between replicates. **d** Graph of select mutants from (**c**). For each mutant, the log2 fraction of normalized reads, normalized to WT, is plotted over time. The slope of a linear regression represents the selection coefficient (SC). Data points are an average of $n = 3$ biological replicates. **e, f** Box and whisker plots showing the selection coefficients of groups of mutants with the indicated number of phosphomimetic mutations (**e**), or with specific substitutions at the indicated positions (**f**). The black center line indicates the median, boxes indicate the 25th–75th percentiles, black lines represent 1.5 interquartile range (IQR) of the 25th and 75th percentile, black circles represent outliers. Data from $n = 3$ biological replicates are included. **g** Mean selection coefficients of indicated mutants. Error bars show standard deviation from the mean in $n = 3$ biological replicates. Significance was tested using repeated measures one-way ANOVA with Dunnett's multiple comparisons test. Asterisks indicate significantly different from WT (SC = 0), ****$P < 0.0001$, ***$P < 0.0005$, ns non-significant. AAAAAAAA, AAAEAAAA, AAAAEAAA $P < 0.0001$; AAAEEAAA $P = 0.3153$; EEEAAEEE $P = 0.0010$; EEEAEEEE $P = 0.9436$; EEEEAEEE, EEEEEEEE $P < 0.0001$. In all panels, the color gradient from blue (*hcm1-8A*) to red (*hcm1-8E*) corresponds to increasing numbers of phosphomimetic mutations.

these data revealed that when all eight sites were mutated, glutamic acid substitutions increased fitness while alanine mutations reduced fitness at any site. However, mutations at two central sites, T460 and S471, had a disproportionally large impact on fitness. In fact, phosphomimetic mutations at only sites T460 and S471 were sufficient to recapitulate WT-like fitness (AAAEEAAA, Fig. 2d, g). Phosphomimetic substitutions at either of these sites individually, as the sole phosphomimetic site, resulted in an intermediate increase in fitness when compared to *hcm1-8A* (AAAEAAAA and AAAAEAAA, Fig. 2g). Conversely, blocking phosphorylation at either site individually reduced fitness when compared to *hcm1-8E*, with T460 having a larger effect (EEEAEEEE and EEEEAEEE, Fig. 2g). Interestingly, although T460 and S471 contributed the greatest amount to fitness, phosphorylation of these sites did not appear to be required, since a mutually exclusive phosphomutant that lacks phosphomimetic mutations at only these sites was also similar in fitness to WT (EEEAAEEE, Fig. 2d, g). Together, these data indicate that both the overall number and the positions of phosphorylated sites can impact cellular fitness.

### Fitness levels predict the transcriptional activation and cell cycle regulatory functions of Hcm1

We next investigated whether the observed cellular fitness phenotypes correlated with Hcm1 activity. We previously found that the fitness phenotypes associated with *hcm1-8A* and *hcm1-8E* correlate with the decreased and increased expression, respectively, of Hcm1 target genes in vivo[25]. In addition, the recruitment of Hcm1-8A to target gene promoters is reduced compared to WT, although it is not eliminated. However, it remains unclear if TAD phosphorylation primarily regulates Hcm1-DNA binding, or if it stimulates its transcriptional activation function in another way. To separate these two possibilities and examine the effects of phosphosite mutations on Hcm1 activity, we utilized a transcriptional activation assay. The previously described Hcm1 TAD excludes the forkhead DNA-binding domain (DBD) but includes all eight C-terminal phosphosites of interest[30]. We fused WT or phosphomutant TAD regions to the C-terminus of the LexA-DBD and expressed these proteins in a reporter strain that contains eight LexA binding sites integrated upstream of the *lacZ* gene[31,32]. β-galactosidase activity assays confirmed that the WT Hcm1 TAD strongly activates transcription (Fig. 3a). Strikingly, expression of *hcm1-8A*, which cannot be phosphorylated, abolished all detectable activity. This demonstrates that phosphorylation is required to activate transcription at a step after Hcm1 binds to DNA. We next examined the activities of several phosphomutants from the screen, to determine if their activities correlated with fitness. The introduction of a single phosphomimetic mutation at either T460 or S471 (AAAEAAAA and AAAAEAAA) restored low levels of activity, and phosphomimetic mutations at both key sites (AAAEEAAA) was sufficient to restore WT-like activity. In addition, the mutant that only lacks phosphomimetic mutations at sites T460 and S471 (EEEAAEEE) displayed activity similar to WT, consistent with its ability to restore near-WT fitness (Fig. 2g). Overall, fitness values determined by Phosphosite Scanning were a good predictor of reporter gene activation. These data suggest that CDK phosphorylation of the TAD tunes Hcm1 activity by recruiting transcriptional activation machinery to stimulate target gene expression.

Next, we determined if the observed fitness phenotypes correlated with Hcm1 function in vivo. Hcm1 regulates several genes that are required for mitotic spindle function and as a result *hcm1Δ* cells are highly sensitive to microtubule poisons[24,33,34]. We therefore tested whether quantitative changes in Hcm1 activity imposed by different combinations of phosphosite mutations imparted similar changes in nocodazole sensitivity. Select phosphomutants were integrated into the genome at the *HCM1* locus and expressed at endogenous levels (Fig. 3b). Doubling times were not significantly changed among the

phosphomutant strains grown in the absence of nocodazole (Fig. 3c). This result was expected since this measurement of proliferation is less sensitive than co-culture assays. However, when cells were grown in a sub-lethal concentration of nocodazole, we observed phosphorylation-dependent sensitivity (Fig. 3d). Although nocodazole treatment slowed the growth of all cells, *hcm1-8A* cells grew approximately three times slower than WT in this condition. Mutants that mimic phosphorylation at either T460 or S471 but lack phosphorylation at all other sites (AAAEAAAA and AAAAEAAA) showed intermediate sensitivity, correlating with the increased fitness imparted by T460 or S471 phosphomimetic mutations. Moreover, mimicking phosphorylation at both T460 and S471 (AAAEEAAA), or at all sites except T460 and S471 (EEEAAEEE), was sufficient to restore nocodazole sensitivity to WT levels. Together, these data demonstrate that the fitness phenotypes of Hcm1 unphosphorylatable and phosphomimetic mutants correlate with their cell cycle regulatory functions in vivo, and that both the position and number of phosphorylations contribute to Hcm1 activity.

### Stabilization of Hcm1 partially mitigates the effects of TAD phosphorylation

In addition to the TAD, CDK also phosphorylates a three-site phosphodegron in the Hcm1 N-terminus that triggers ubiquitin-mediated protein degradation[25]. Blocking phosphorylation of these sites with alanine substitutions (*hcm1-3N*) stabilizes Hcm1 (Supplementary Fig. 5a) and lengthens its expression through the cell cycle[25]. We hypothesized that maintaining low levels of Hcm1 might be important for cells to respond to dynamic phosphorylation of the TAD, and as a result, phosphomimetic mutations might not enhance cellular fitness to the same extent in the *hcm1-3N* mutant, in which Hcm1 levels are stabilized throughout the cell cycle.

To test this hypothesis, we performed pairwise co-culture assays to evaluate the fitness effects of unphosphorylatable and phosphomimetic mutations in the TAD when Hcm1 is stabilized. First, we confirmed that *hcm1-3N* increases fitness compared to WT, as previously reported[26] (Fig. 4a). Next, we compared *hcm1-3N* to *hcm1-3N8A* and found that blocking phosphorylation of the TAD also reduced fitness when Hcm1 was stabilized (Fig. 4b). This result is consistent with the finding that *hcm1-8A* is unable to activate transcription (Fig. 3a). However, in contrast to the fitness benefit of *hcm1-8E* compared to WT (Fig. 1e), *hcm1-3N8E* did not increase cellular fitness compared to *hcm1-3N* (Fig. 4c), in support of the hypothesis that TAD phosphorylation is less beneficial when Hcm1 is stabilized. These data argue that the balance between protein levels and activating phosphorylations is important for optimal regulation of Hcm1 activity.

To investigate this result more comprehensively, we performed a second Phosphosite Scanning screen utilizing a phosphomutant plasmid library in which the A/E mutations were introduced into an *HCM1* allele that contained stabilizing mutations in the N-terminus (referred to as the 3N A/E library, Fig. 4d). This screen was performed in triplicate and analyzed for cellular fitness outcomes. However, since all *HCM1* alleles that were screened included stabilizing mutations, the calculated selection coefficients were normalized to the *hcm1-3N* allele (not WT *HCM1*). We found that in the presence of stabilizing mutations, fitness also increased with the number of phosphomimetic mutations in the TAD (Fig. 4e). Interestingly, in contrast to what we observed in pairwise experiments, several mutants with large numbers of phosphomimetic substitutions, including *hcm1-3N8E*, were slightly more fit than *hcm1-3N* when the library was screened in bulk (Supplementary Fig. 5b). However, for all phosphomimetic mutants, increases in fitness in the 3N background were reduced relative to the WT (non-3N) background (Supplementary Fig. 5c). For example, the average selection coefficient of the *hcm1-8E* mutant was 0.007 in the 3N A/E screen and 0.016 in the A/E screen. This suggests that our sequencing-based screens may be more sensitive than pairwise competition assays.

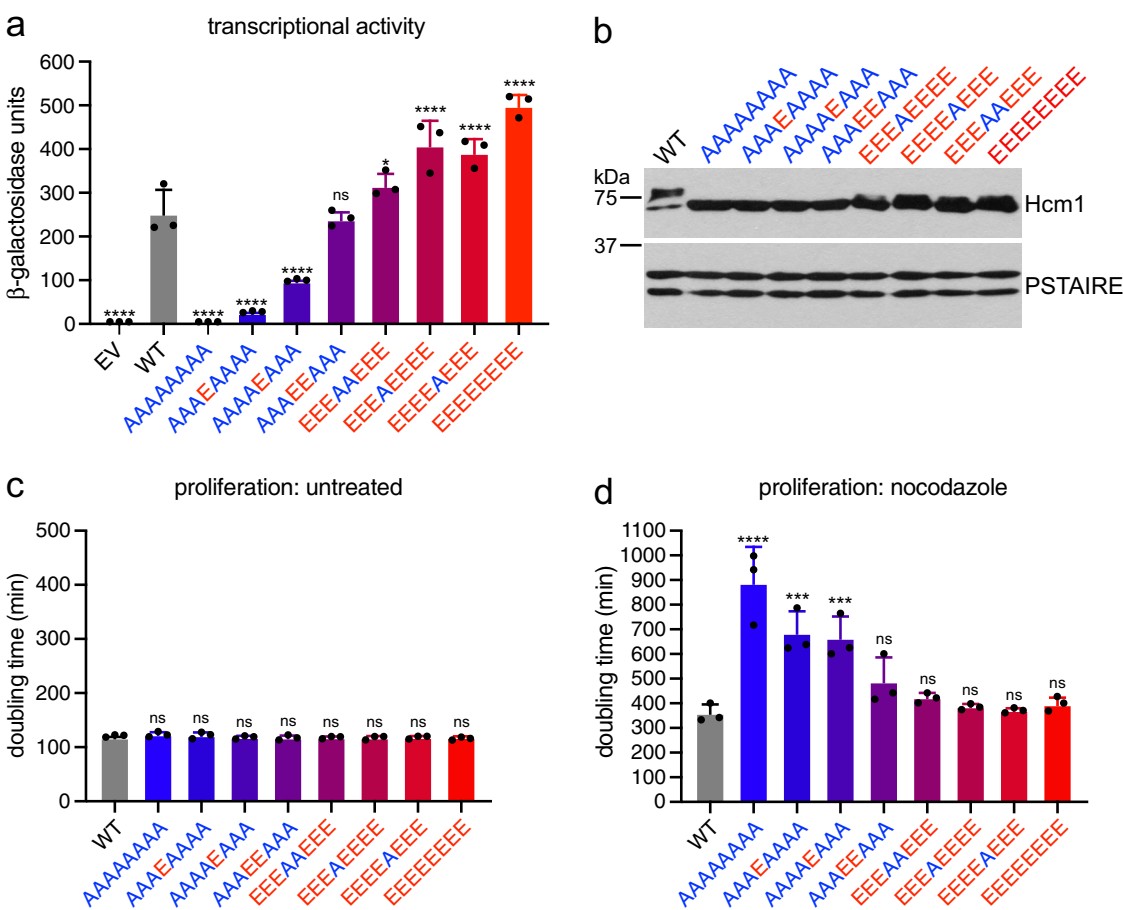

**Fig. 3 | Fitness values predict the transcriptional activation and cell cycle regulatory functions of Hcm1. a** Transcriptional activation assay of LexA-DBD fused to the Hcm1 TAD with the indicated phosphosite mutations, and WT or empty vector (EV) controls. Black represents EV, gray represents WT, color gradient from blue (*hcm1-8A*) to red (*hcm1-8E*) corresponds to increasing number of phospho-mimetic mutations. Error bars represent standard deviation, shown is an average of $n = 3$ replicates. Significance was tested using repeated measures one-way ANOVA with Dunnett's multiple comparisons test. Asterisks indicate significantly different from WT, ****$P < 0.0001$, *$P < 0.05$, ns = non-significant. EV, AAAAAAAA, AAAEAAAA, AAAAEAAA $P < 0.0001$; AAAEEAAA $P = 0.9922$; EEEAAEEE $P = 0.0484$; EEEAEEEE, EEEEAEEE, EEEEEEEE, $P < 0.0001$. Supplementary Fig. 4a confirms that the fusion proteins are expressed at similar levels. **b** Western blot of the indicated Hcm1 proteins expressed from the endogenous *HCM1* locus. Hcm1 was detected using an antibody against the C-terminal 3V5 tag, PSTAIRE shown as a loading control. Representative blots from $n = 3$ replicates are shown. **c, d** Doubling times of strains from (**b**) growing at 30 °C in rich medium (**c**) or in rich medium with 5 μg/mL nocodazole (**d**). Gray represents WT, color gradient from blue (*hcm1-8A*) to red (*hcm1-8E*) corresponds to increasing numbers of phosphomimetic mutations. Error bars represent standard deviation, shown is an average of $n = 3$ replicates. Significance was tested using repeated measures one-way ANOVA with Dunnett's multiple comparisons test. Asterisks indicate significantly different from WT, ****$P < 0.0001$, ***$P < 0.0005$, ns = non-significant. **d** AAAAAAAA $P < 0.0001$; AAAEAAAA $P = 0.0002$; AAAAEAAA $P = 0.0003$; AAAEEAAA $P = 0.1666$; EEEAAEEE $P = 0.7932$; EEEAEEEE $P = 0.9971$; EEEEAEEE $P = 0.9997$; EEEEEEEE $P = 0.9886$.

We next compared the importance of individual phosphosites in the 3N background and found that, like the A/E phosphomutants, mimicking phosphorylation at any site increased fitness, with sites T460 and S471 having the largest impact (Fig. 4f). Interestingly, when we compared the results of the A/E and 3N A/E screens, we found that the fitness of both phosphomimetic and phosphodeficient mutants were lower in the 3N background, such that the positive effects of most phosphomimetic mutations were partially blunted and the negative effects of most phosphodeficient mutations were exacerbated (Supplementary Fig. 5b). As a result, a larger fraction of 3N A/E mutants had a negative effect on fitness compared to A/E mutants (Fig. 4g). This effect was strongest among the least fit mutants. Together, these data support the model that increasing the levels of Hcm1 reduces the importance of TAD phosphorylation for Hcm1 activity and fitness.

**Quantifying fitness of mutants that incorporate WT phosphosites**

After identifying phosphomimetic mutations that activate Hcm1, we wanted to investigate if these sites are phosphorylated in vivo. To accomplish this, we performed two additional Phosphosite Scanning screens to interrogate phosphomutant plasmid libraries that incorporate WT sites. We reasoned that comparing the fitness outcomes of mutants that incorporate WT sites and analogous mutants where the phosphorylation status is fixed (A/E library) would allow us to infer which sites are phosphorylated in vivo.

In the first screen, we constructed a library in which each phosphoacceptor site was mutated to alanine or left as the WT S-P or T-P site in all possible combinations (referred to as the WT/A library, Fig. 5a). If all sites contribute to Hcm1 activity in vivo, our expectation was that all phosphomutants in this library should display either WT-like or deleterious fitness. Consistent with this prediction, examination of each mutant's abundance over time confirmed that all mutants displayed cellular fitness that was equal to or worse than WT (Supplementary Fig. 6a). In general, fitness increased with the total number of available WT sites (Fig. 5b), and a WT S/T-P site at any location increased fitness (Fig. 5c). These data suggest that all eight sites are phosphorylated in vivo and contribute to Hcm1 activation.

In the second screen, we constructed a library in which each site was either WT or a phosphomimetic mutation (referred to as the WT/E library, Fig. 5d). Based on our previous observations, we expected to

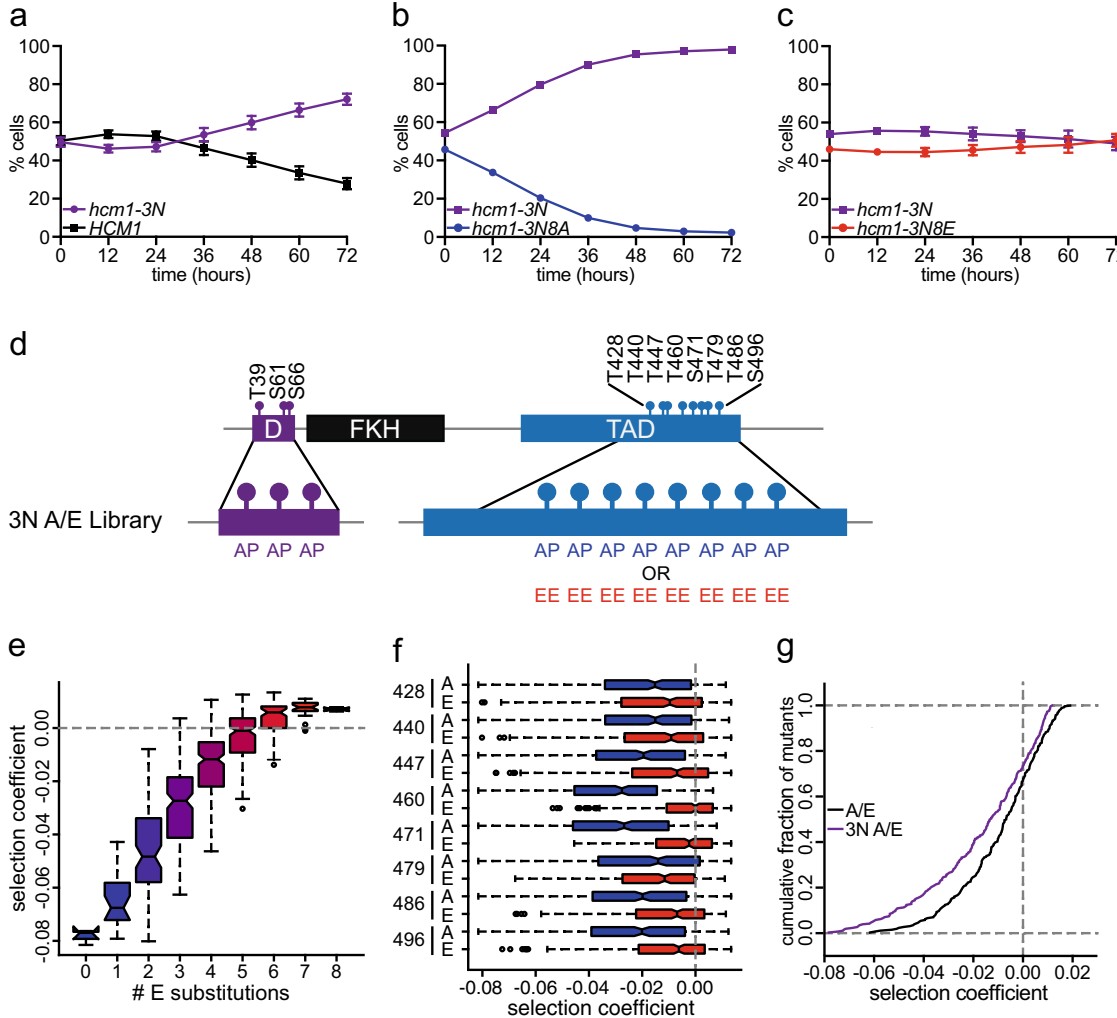

**Fig. 4 | Stabilization of Hcm1 partially mitigates the effects of TAD phosphorylation. a–c** Strains expressing the indicated *HCM1* alleles from plasmids were co-cultured and the percentage of cells expressing each allele was determined at the timepoints indicated. In total, 5000 cells were analyzed from each timepoint. Shown is an average of *n* = 3 experiments, error bars represent standard deviations. Black represents cells expressing WT *HCM1* (**a**), purple represents cells expressing *hcm1-3N* (**a–c**), blue represents cells expressing *hcm1-3N8A* (**b**), and red represents cells expressing *hcm1-3N8E* (**c**). **d** Schematic showing the design of the 3N A/E phosphomutant library. D indicates the phosphodegron consisting of three S/T-P sites that have been mutated to A-P (3N). The eight S/T-P sites in the TAD have been mutated to either A-P or E-E in all 256 possible combinations. **e, f** Box and whisker plots showing the median selection coefficients of groups of mutants with the indicated number of phosphomimetic mutations (**e**), or with specific substitutions at the indicated positions (**f**). Color gradient from blue (*hcm1-3N8A*) to red (*hcm1-*

*3N8E*) corresponds to increasing numbers of phosphomimetic mutations within the Hcm1 TAD (**e**), Blue corresponds to all mutants with A-P at the indicated site, red corresponds to all mutants with E-E at the indicated site (**f**). The black center line indicates the median, boxes indicate the 25th–75th percentiles, black lines represent 1.5 IQR of the 25th and 75th percentile, black circles represent outliers. Shown is all data from *n* = 3 biological replicates. Supplementary Fig. 3b shows correlations between replicates; Supplementary Fig. 5 shows elevated expression of Hcm1-3N and heatmaps comparing A/E and 3N A/E screens. **g** Cumulative frequency plot showing the cumulative fraction of mutants that display selection coefficients that are less than or equal to the indicated values. Black represents the A/E library, purple represents the 3N A/E library. Note that selection coefficients in the A/E library are calculated with relative to WT *HCM1* and selection coefficients in the 3N A/E library are calculated relative to *hcm1-3N*. Shown is data from *n* = 3 biological replicates.

see a fitness benefit in highly phosphomimetic mutants. Indeed, we found that selection coefficients tended to increase with the number of phosphomimetic sites, consistent with what we observed in previous screens (Fig. 5e). In fact, the median selection coefficient of all groups of WT/E mutants with one or more phosphomimetic sites was higher than that of WT.

Next, we wanted to investigate the impact of phosphomimetic substitutions in more detail. Because phosphomimetic mutations at any site were beneficial compared to alanine substitutions, and all phosphosites in the library are WT or phosphomimetic, we expected that all individual phosphomutants in this library would exhibit fitness that is equal to or better than WT. Remarkably, several mutants were much less fit than WT (Supplementary Fig. 6b). Examination of these mutants suggested that fitness phenotypes might be driven by E

substitutions at one or more of the first three phosphosites (Supplementary Fig. 6c). To investigate this possibility quantitatively, we compared median selection coefficients of mutants based on their genotype at each site. Indeed, in contrast to what we observed in A/E and 3NA/E screens, we found that mutation of the first two sites to glutamic acid reduced fitness (T428 and T440, Fig. 5f). These data suggest that phosphomimetic mutations at some phosphosites are detrimental for fitness when WT sites are also present.

## Phosphorylation of sites 460 and 471 depends on upstream CDK sites

To investigate why phosphomimetic mutations at T428 and T440 led to a reduction in fitness, we compared the selection coefficients of several phosphomutants from the WT/E screen. Mutation of either

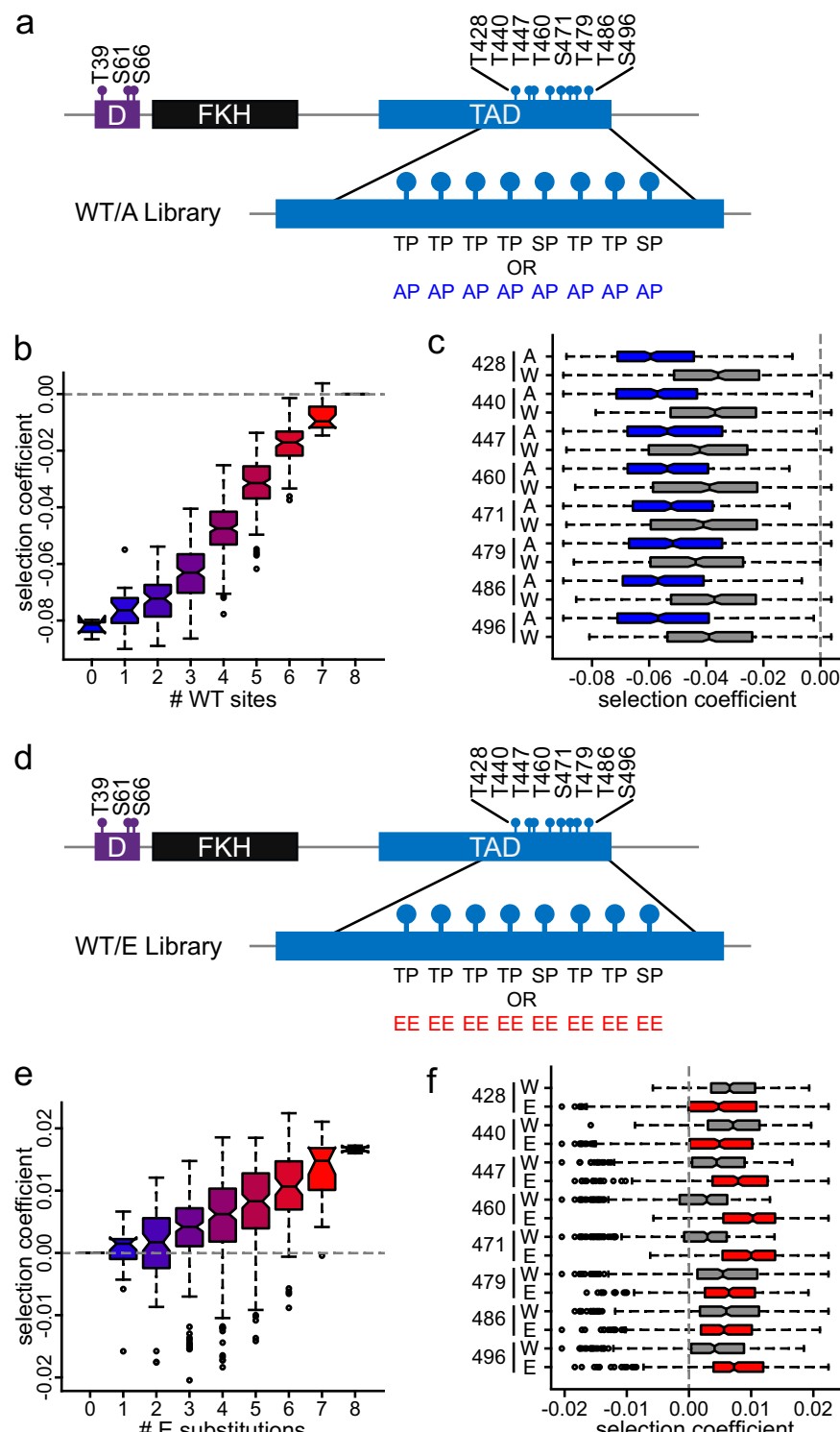

T428 or T440 to glutamic acid resulted in a modest, but not significant, reduction in fitness with respect WT (EWWWWWWW and WEWWWWWW, Fig. 6a). This effect was strengthened and significant when both sites were mutated simultaneously (EEWWWWWW). Similarly, mutation of the first three positions resulted in a loss of fitness of approximately the same magnitude (EEEWWWWW). These data raised the possibility that WT sites C-terminal to these mutations, including the most impactful sites for Hcm1 activity, T460 and S471, may not be phosphorylated in these mutants. Interestingly, adding a fourth phosphomimetic mutation at position 460 restored fitness to WT levels (EEEEWWWW), in support of the possibility that T460 is not

phosphorylated if the three N-terminal sites within the TAD are changed to E. This mutant was similar in fitness to a mutant that has a single phosphomimetic substitution at T460 (WWWEWWWW), suggesting that the primary function of the first three sites is to promote phosphorylation of more C-terminal sites in the TAD.

One possibility is that although E substitutions mimic the charge of the phosphate and are sufficient for TAD activity, they may interfere with phosphorylation of more C-terminal sites because they are structurally dissimilar to a phosphate. If this is the case, then mutation of these sites to A would also be predicted to disrupt phosphorylation of more C-terminal sites. We examined several key mutants from the A/

**Fig. 5 | Quantifying fitness of mutants that incorporate WT phosphosites.**
**a** Schematic showing the design of the WT/A plasmid library. S/T-P sites within the TAD were either left intact or mutated to A-P in all possible combinations (256). **b**, **c** Box and whisker plots showing the selection coefficients of groups of mutants with the indicated number of WT (W) S/T-P sites (**b**), or with specific substitutions at the indicated positions (**c**). Color gradient from blue (hcm1-8A) to red (HCM1) corresponds to increasing numbers of WT sites (**b**). Blue represents all mutants with an A-P mutation at the indicated site, gray represents all mutants with wild-type sequences at the indicated site (**c**). The black center line indicates the median, boxes indicate the 25th–75th percentiles, black lines represent 1.5 IQR of the 25th and 75th percentile, black circles represent outliers. Shown is data from n = 3 bio-logical replicates. See Supplementary Fig. 3c for correlations between replicates and Supplementary Fig. 6a for heatmap of screen results. **d** Schematic showing the

design of the WT/E plasmid library. S/T-P sites in the TAD were either left intact or mutated to E-E in all possible combinations (256). **e**, **f** Box and whisker plots showing the selection coefficients of groups of mutants with the indicated number of phosphomimetic mutations (**e**), or with specific substitutions at the indicated positions (**f**). Color gradient from blue (HCM1) to red (hcm1-8E) corresponds to increasing numbers of phosphomimetic mutations (**e**). Gray represents all mutants with WT sequence at the indicated site, red represents all mutants that have an E-E mutation at the indicated site (**f**). The black center line indicates the median, boxes indicate the 25th–75th percentiles, black lines represent 1.5 IQR of the 25th and 75th percentile, black circles represent outliers. Shown is data from n = 3 biological replicates. See Supplementary Fig. 3d for correlations between replicates and Supplementary Fig. 6b, c for heatmaps of screen results.

E, WT/A, and WT/E screens and found two comparisons that supported this model (Fig. 6b). First, we found in our initial screen that hcm1-8A has a severe fitness defect, and that mimicking phosphorylation at only sites T460 and S471 (AAAEEAAA) was sufficient to restore WT fitness (Figs. 2g and 6b). We reasoned that if sites T460 and S471 are phosphorylated in vivo then a similar mutant with WT sequence at only T460 and S471 (AAAWWAAA) should also show WT-like fitness. How-ever, this mutant showed a fitness defect approximately equal to hcm1-8A, suggesting the sites T460 and S471 are not phosphorylated when all other phosphosites are mutated to alanine (Fig. 6b, compare three leftmost bars). These results were supported by transcriptional acti-vation assays, which demonstrated that the AAAWWAAA mutant is unable to activate transcription (Fig. 6c). Second, our initial screen revealed that the fully phosphomimetic mutant (EEEEEEEE, hcm1-8E) displayed the largest fitness increase. Again, if T460 and S471 are phosphorylated in vivo, we would expect that a mutant with WT sequence at these positions (EEEWWEEE) would display similar fitness and activity. However, we found that this mutant exhibited reduced fitness compared to hcm1-8E and was identical to the corresponding alanine mutant (EEEAAEEE) (Fig. 6b, compare three rightmost bars). These two mutants (EEEWWEEE and EEEAAEEE) also showed similar transcriptional activity, which was reduced compared to hcm1-8E (Fig. 6c). Together, these results suggest that mutation of the first three phosphosites to either A or E impairs phosphorylation of T460 and S471, which in turn reduces Hcm1 activation.

Next, we sought to determine whether T460 and S471 are phos-phorylated in the context of the WT protein in vivo. Phosphomutants that only lack phosphoacceptor sites at T460 and/or S471 displayed a modest reduction in fitness, in support of this possibility (WWWAAWWW, WWWAWWWW, WWWWAWWW, Fig. 6b). To inves-tigate phosphorylation more directly, we examined the migration of Hcm1 mutants using Phos-tag gels—which increase the mobility shift of proteins during SDS-PAGE in proportion to their phosphorylation levels—followed by Western blotting[35]. As previously shown[26], most of the WT Hcm1 protein migrates as a single band near the top of a Phos-tag gel (Fig. 6d). In contrast, a mutant that lacks all CDK consensus sites (15A) migrates as three bands near the bottom of the gel, confirming that the WT protein is highly phosphorylated on S/T-P sites. Impor-tantly, a mutant with alanine substitutions at T460 and S471 (WWWAAWWW) migrates faster than the WT protein, and the corre-sponding single mutants are intermediate, providing strong evidence that T460 and S471 are phosphorylated in vivo (Fig. 6d, rightmost four lanes). Notably, there was no evidence of phosphorylation at T460 and/or S471 when all other sites were mutated to alanine. Mutants that are WT at only T460, S471, or both positions migrate identically to the Hcm1-8A mutant that lacks all phosphosites in the TAD, supporting the possibility that mutation of the first three positions prevents phos-phorylation at more C-terminal sites (Fig. 6d, leftmost five lanes). Similar results were observed when phosphorylation was examined in an Hcm1 protein that lacked all S/T-P sites except for those in the TAD (7N, Fig. 6e), and in LexA-Hcm1 TAD fusion proteins (Supplementary

Fig. 4d). Together, these data demonstrate that T460 and S471 are phosphorylated in vivo and suggest that their phosphorylation depends upon the presence of WT sequence at N-terminal CDK sites.

### Phosphorylation of sites 460 and 471 depends on Cks1
Since phosphorylation of T460 and S471 requires N-terminal CDK sites, we wondered if the phosphoadaptor protein Cks1 is involved in Hcm1 TAD phosphorylation. Cks1 forms a complex with cyclin-CDK and recruits the complex to already phosphorylated priming sites to trigger a processive multisite phosphorylation cascade in the N- to C-terminal direction[27,36–38]. Regulation by Cks1 requires a phosphory-lated threonine priming site that is 12–30 residues upstream of addi-tional CDK consensus sites. Interestingly, the spacing of CDK phosphosites in the Hcm1 TAD makes it a good candidate to be regulated by Cks1 (Fig. 7a). In addition, Cks1 appeared to facilitate processive Hcm1 phosphorylation in an in vitro kinase assay[27]. How-ever, it is unknown if and how Cks1 regulates Hcm1 phosphorylation in vivo. Based on our phosphomutant analysis, we hypothesized that one or more of the first three CDK sites in the Hcm1 TAD acts as a priming site for Cks1, which in turn promotes C-terminal phosphor-ylation that contributes to Hcm1 activation.

Since Cks1 binds to phosphorylated threonines, but not phos-phorylated serines[27,38], we examined Cks1 involvement in Hcm1 phos-phorylation by changing T-P motifs to S-P motifs within the Hcm1 TAD. These T to S mutants can still be phosphorylated by CDK but cannot serve as priming sites for Cks1. Strikingly, mutation of the first three or four threonine residues to serine reduced the extent of Hcm1 phos-phorylation, as evident by a ladder of intermediate migrating bands that were not observed with the WT protein (SSSSWWWW and SSSWWWWW, Fig. 7b). Next, we tested if the serine substitutions resulted in a corresponding reduction in transcriptional activation by Hcm1. A modest, but significant, reduction in activity was observed in both mutants compared to WT (Fig. 7c). We also considered the pos-sibility that T460 could serve as a priming site to facilitate phosphor-ylation of S471 and/or downstream sites. There was no evidence of a reduction in phosphorylation when T460 alone was changed to serine (WWWSWWWW, Fig. 7b), however the activity of this mutant was reduced, suggesting that it may impact phosphorylation of one more C-terminal sites despite having a minimal impact on protein migration in a Phos-tag gel. Together, these results are consistent with a model in which the N-terminal threonine residues within the TAD can serve as Cks1 priming sites to facilitate the phosphorylation of downstream sites and enhance Hcm1 activity. These data further reveal that phos-phomimetic mutations can substitute for bona fide phosphorylation for TAD activation, but not Cks1 recruitment.

## Discussion
### Multisite phosphorylation by CDK activates Hcm1
Here, we describe a high-throughput selection approach, Phospho-site Scanning, that can be used to determine the contributions of individual phosphosites within a multisite phosphorylated domain.

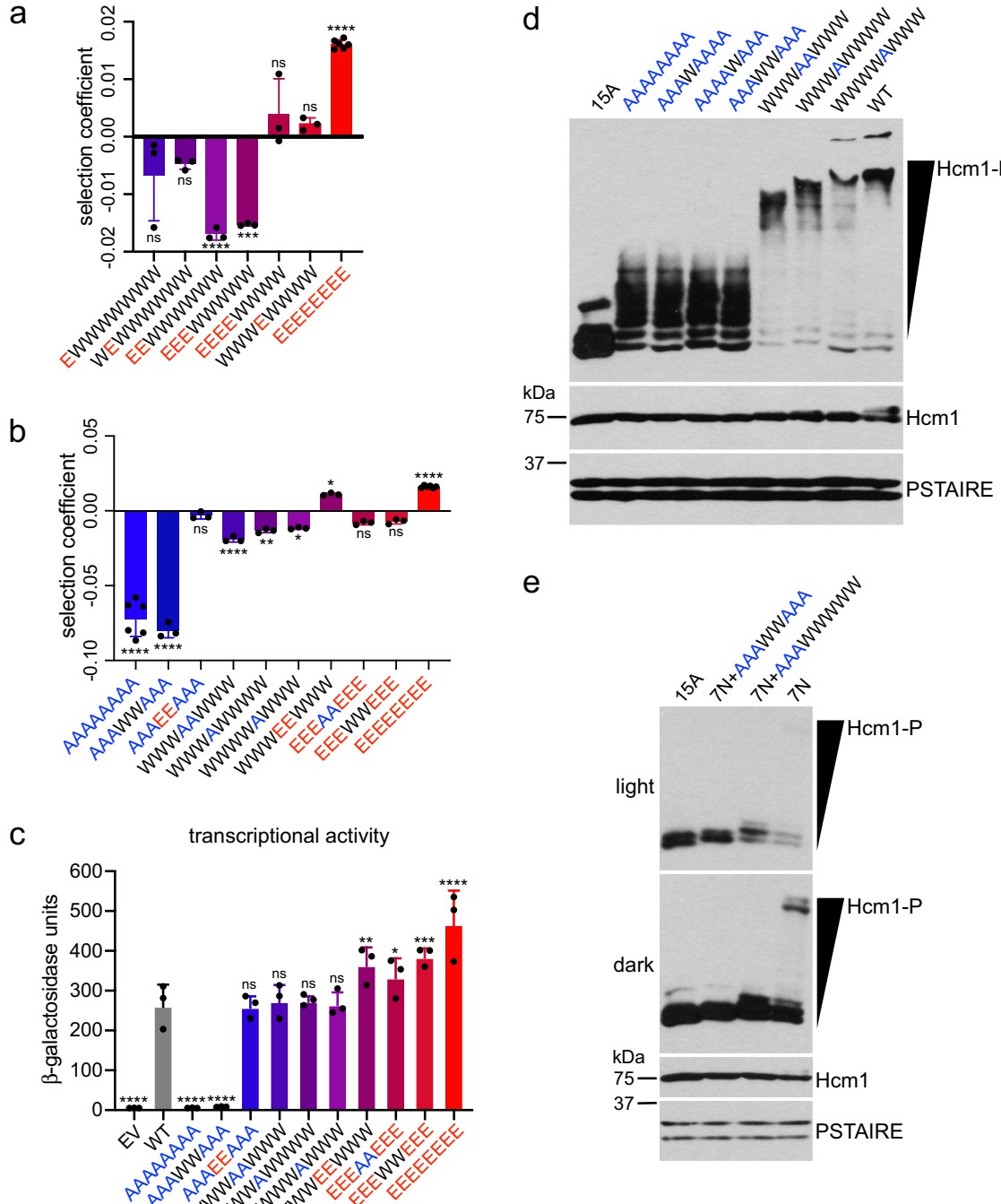

**Fig. 6 | Phosphorylation of sites 460 and 471 depends on upstream CDK sites.**
**a**, **b** Selection coefficients of the indicated mutants from the WT/E screen (**a**) or
multiple screens (**b**). Error bars represent standard deviation, shown is an average
of $n = 3$ biological replicates for all mutants except AAAAAAAA and EEEEEEEE,
which are an average of $n = 6$ biological replicates. Significance was tested using
ordinary one-way ANOVA with Dunnett's multiple comparisons test. Asterisks
indicate significantly different from WT (SC = 0), ****$P < 0.0001$, ***$P < 0.0005$,
**$P < 0.005$, *$P < 0.05$, ns non-significant. Exact $P$ values are included in the Source
Data. **c** Transcriptional activation assay of fusion constructs containing the LexA-
DBD fused to the Hcm1 TAD with the indicated phosphosite mutations, or WT or EV
controls. Error bars represent standard deviations, shown is an average of $n = 3$
replicates. Significance was tested using repeated measures one-way ANOVA with
Dunnett's multiple comparisons test. Asterisks indicate significantly different from

WT, ****$P < 0.0001$, ***$P < 0.0005$, **$P < 0.005$, *$P < 0.05$, ns non-significant. Exact $P$
values are included in the Source Data. Supplementary Fig. 4b confirms that all
fusion proteins are expressed at similar levels. The color gradient from blue (hcm1-
8A) to red (hcm1-8E) corresponds to increasing numbers of phosphomimetic
mutations (**a**–**c**). **d** Phos-tag western blot of the indicated 3V5-tagged Hcm1 pro-
teins (top panel) and standard western blots showing expression of the indicated
proteins (lower panels). PSTAIRE is shown as a loading control. Representative
images from $n = 3$ biological replicates are shown. **e** Phos-tag western blots, as in (**d**)
comparing Hcm1 mutants with A-P mutations in all S/T-P sites, except for the
indicated sites in the TAD. The previously described 7N mutant[25] has the seven
N-terminal S/T-P sites (all sites except those in the TAD) mutated to A-P, sites in the
TAD have the indicated mutations. Representative images from $n = 4$ biological
replicates are shown.

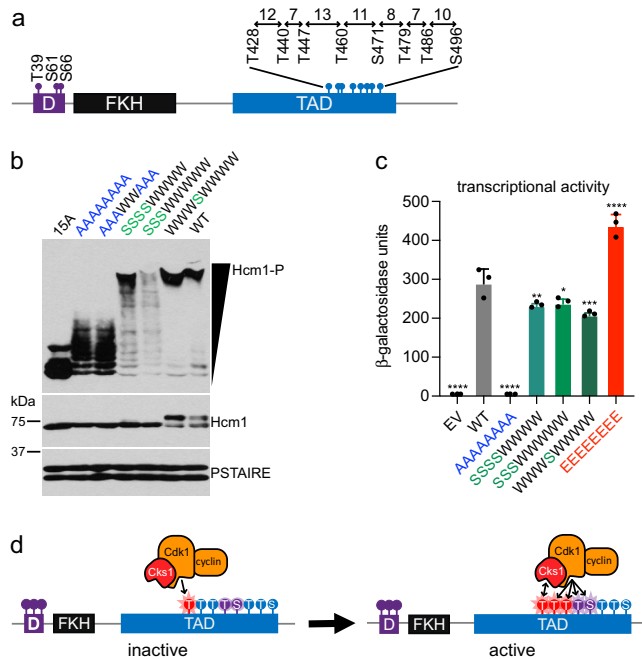

**Fig. 7 | Phosphorylation of sites 460 and 471 depends on Cks1. a** Schematic showing the spacing between phosphosites in the TAD. **b** Phos-tag western blot showing phosphorylation of the indicated Hcm1 proteins (top panel) and standard western blots showing expression of the indicated proteins (lower panels). Hcm1 was detected with antibody against the 3V5 tag, PSTAIRE is shown as a loading control. Representative images from $n = 3$ biological replicates are shown. **c** Transcriptional activation assay of constructs containing the LexA-DBD fused to the Hcm1 TAD with the indicated phosphosite mutations or WT or empty vector (EV) controls. Black represents EV, gray represents WT, blue represents *hcm1-8A*, red represents *hcm1-8E*, and shades of green represent mutants with potential T priming sites mutated to S. Error bars represent standard deviation, shown is an average of $n = 3$ replicates. Significance was tested using repeated measures one-way ANOVA with Dunnett's multiple comparisons test. Asterisks indicate significantly different from WT, ****$P < 0.0001$, ***$P < 0.0005$, **$P < 0.005$, *$P < 0.05$. EV, AAAAAAAA $P < 0.0001$; SSSSWWWW $P = 0.0064$; SSSWWWWW $P = 0.0154$; WWWSWWWW $P = 0.0005$; EEEEEEEE $P < 0.0001$. Supplementary Fig. 4c confirms that all fusion proteins are expressed at similar levels. **d** Model depicting the role of Cks1 in multisite phosphorylation of the Hcm1 TAD.

We used this approach to dissect the rules of phosphorylation within a model CDK-regulated domain, the TAD of the transcriptional activator Hcm1. First, by screening a library with all possible combinations of unphosphorylatable and phosphomimetic mutations, we were able to elucidate which combinations of phosphorylations can activate Hcm1. This analysis revealed that Hcm1 activity is tunable and increases with the number of phosphomimetic mutations (Fig. 2e). However, not all sites are equivalent, and some contribute a greater amount to activation (Fig. 2f). Remarkably, WT-like fitness and Hcm1 activity can be achieved by non-overlapping patterns of phosphomimetic mutations (Figs. 2g and 3a, d). Although this screen revealed that it is possible to achieve activity in many ways, it did not elucidate which sites contribute to activation in the context of the WT protein. For this reason, we also performed screens that incorporated WT sites. Screening of the WT/A library revealed that mutation of any site to alanine reduces fitness, suggesting that all sites are phosphorylated to some extent in vivo (Fig. 5c). Screening of the WT/E library revealed more complex regulation. Phosphomimetic mutation of sites at the N-terminal end of the cluster prevented phosphorylation of more C-terminal sites (Fig. 5f). Our results suggest that the most N-terminal sites within the TAD both contribute to transcriptional activation and perform a structural role as docking sites for the CDK-associated phosphobinding protein Cks1.

Together, these screens suggest a model for how the Hcm1 TAD is regulated by CDK in vivo (Fig. 7d). Our data suggest that CDK first phosphorylates an upstream threonine consensus site within the TAD region. This phosphorylated threonine then serves as a docking site for Cks1. Cks1 binds the phosphorylated priming site, triggering a multi-site phosphorylation cascade in the N- to C-terminal direction. This allows for the rapid phosphorylation of two key sites, T460 and S471, which have the greatest impact on Hcm1 transcriptional activity, resulting in the expression of Hcm1 target genes. Notably, a comparison of Hcm1 sequences among budding yeasts revealed that the five N-terminal phosphosites in the TAD are more highly conserved that the three C-terminal sites (Supplementary Fig. 7a), consistent with our model that the N-terminal and central sites are critical for Hcm1 activation.

Although Cks1 has been shown to enhance the phosphorylation of Hcm1 in vitro[27], our results provide the first evidence that Hcm1 phosphorylation is stimulated by Cks1 in vivo. To facilitate processive phosphorylation, Cks1 binds to a phosphorylated threonine that is 12–30 amino acids upstream of additional CDK sites. Based on their spacing, any of the first four CDK sites in the Hcm1 TAD could act as priming sites for one or more downstream sites (Fig. 7a). In fact, all threonine residues in the TAD, except for T440, were previously identified as putative Cks1-binding sequences[38]. Our expectation was that Cks1 binding at the N-terminal end of the cluster would have a greater impact on phosphorylation, since it could initiate a cascade of phosphorylation that would lead to rapid phosphorylation of T460 and S471 and activation of the protein. In support of this model, the extent of Hcm1 phosphorylation was reduced when the three N-terminal threonines within the TAD were mutated to serine, whereas the T460S mutation alone had less of an effect (Fig. 7b). Interestingly, although the T460S mutation alone did not impair Hcm1 phosphorylation as visualized by Phos-tag western blot, the activity of this mutant was significantly reduced compared to WT (Fig. 7b, c). Therefore, it is possible that T460S does impair phosphorylation of one or more downstream sites, but T460 phosphorylation may impact fewer sites than T428/T440/T447.

Phosphorylation of the Hcm1 TAD activates the expression of Hcm1 target genes in vivo. Our data indicate that TAD phosphorylation stimulates transcriptional activation through a charge-based mechanism that is independent of DNA binding (Fig. 3a). A likely possibility is that phosphorylation causes a conformational change that promotes binding to a coactivator. Interaction with a coactivator could stabilize the Hcm1-DNA interaction, and this might explain why the unphosphorylatable *hcm1-8A* protein exhibits decreased recruitment, and *hcm1-8E* displays enhanced recruitment, to target gene promoters in vivo[25]. Our results suggest that a balance between the overall number and precise positioning of phosphorylated residues within the TAD determines cofactor binding. Interestingly, the key sites T460 and S471 are centrally located within the phosphosite cluster and are in a region that is predicted to be more structured than the surrounding sequence (Fig. 1a). The TAD structure is not predicted to change markedly in the presence of phosphosite mutations (Supplementary Fig. 7b). However, analysis of yeast and human TADs have shown that negative charge is important in the TAD to keep critical aromatic resides solvent exposed, which enables binding to coactivators[39,40]. Phosphorylation of T460 and S471 may increase the negative charge in this critical region, leading to coactivator recruitment. This possibility is also consistent with the fact that a mutant that has phosphomimetic mutations at all sites except T460 and S471 (EEEAAEEE) can also support transcriptional activity, since an increased amount of negative charge that is less optimally placed could be sufficient for cofactor recruitment. We anticipate that this mechanism of regulation may be common among multisite phosphorylated substrates: that regulation can occur via different combinations of phosphosites, but that evolution has selected for sequence

contexts that promote rapid phosphorylation and switch-like regulation.

### Decoding multisite phosphorylation by phosphosite scanning

Historically, determining which phosphosites within a multiphosphorylated domain impact protein function has been challenging, since it requires making large numbers of mutant alleles and screening them individually. Phosphosite Scanning offers several benefits over traditional approaches. Since all possible combinations of alleles can be generated and screened in bulk, it is both comprehensive and rapid. Moreover, as is evident from our data, fitness readouts are very sensitive and can detect small differences in activity among mutants. The ideal substrate for this type of screen is a protein like Hcm1, in which eliminating and mimicking phosphorylation lead to opposing effects on protein function and cellular fitness. This approach can easily be applied to any phosphorylated protein that is regulated by multisite phosphorylation and demonstrates phosphorylation-dependent fitness effects. Phosphosite Scanning could also be adapted for substrates that do not meet these criteria. Instead of measuring cellular fitness as a readout, protein function could be measured in another way. For example, screens could be developed that utilize GFP as a readout, and mutants with different effects on activity determined by sorting cells based on GFP expression. This type of approach has been successful in combination with traditional deep mutational scanning[39,40].

One consideration when designing a Phosphosite Scanning screen is that the functional consequence of phosphorylation is not always replicated by phosphomimetic mutations. Substitution of phosphoacceptor sites with two glutamic acids replicates the change of charge caused by phosphorylation. This can mimic the outcome of phosphorylation when the change of charge of the substrate region is important for protein function[18]. However, there are instances when the addition of a phosphate group is required, and the effect cannot be replicated by charge alone. This is often the case when phosphorylation regulates a protein-protein interaction. For example, the degradation of many substrates of SCF ubiquitin ligases depends on substrate phosphorylation to form a phosphodegron motif that can interact with an F-box protein[16,17]. Since glutamic acid substitutions are structurally dissimilar to a phosphate, they are unable to form an interaction motif. Similarly, interaction between a CDK substrate and the processivity factor Cks1 depends on a phosphorylated threonine priming site within the substrate[27,38]. As our data shows, phosphomimetic substitutions at the most N-terminal threonine residues in the Hcm1 TAD prevented phosphorylation of more C-terminal WT sites, suggesting that they do not interact with Cks1. Despite these limitations, screens of phosphomimetic mutations can be informative for many targets. Alternatively, variations on Phosphosite Scanning that only interrogate unphosphorylatable substitutions could also be used to determine which sites are required for a functional output.

### The importance of Hcm1 phosphoregulation

Hcm1 is an ideal model substrate to elucidate the principles of regulation by CDK because the TAD shares many features of domains that are multisite phosphorylated[3], and its phosphorylation leads to clear fitness phenotypes that correlate extremely well with protein function. However, the fact that cells expressing the fully phosphomimetic mutant (hcm1-8E) are more fit than WT cells suggests that not all sites are phosphorylated at any one time and raises the interesting question of why Hcm1 phosphoregulation has evolved. A likely possibility is that in some conditions having too much Hcm1 activity is detrimental, so it is beneficial to keep its levels low and rapidly toggle activity on and off via phosphorylation, as needed. In support of this model, activating phosphates in the TAD are specifically removed by the phosphatase calcineurin in response to stress[26]. We show here that phosphorylation-

mediated degradation is important to sensitize Hcm1 to changes in TAD phosphorylation, since phosphomimetic mutations in the TAD do not provide the same fitness advantage when N-terminal phosphosites are mutated and the protein is stabilized (Fig. 4). The complex phosphoregulation of Hcm1 activity suggests that it is a critical regulator that modulates fitness in different environments. Future studies examining the consequence of Hcm1 phosphomutants in different environments should shed light on this possibility.

## Methods

### Yeast strains and plasmids

All strains are described in Supplementary Table 1. Genetic manipulations were performed using standard methods[41,42]. Strains were grown in rich medium (YM-1) or synthetic medium lacking a single amino acid with 2% dextrose or galactose, where indicated, at 30 °C. Strains with *HCM1* mutations integrated into the genome were generated from synthesized gene fragments (Invitrogen) that were cloned into pFA6a-3V5-KanMX6 using the NEBuilder HiFi DNA Assembly kit (New England Biolabs). All genomic mutations were confirmed by sequencing. *GAL1p-HCM1* was introduced into strains for pairwise competition assays by genetic cross with previously characterized strains expressing WT or non-fluorescent GFP[43].

All plasmids are listed in Supplementary Table 2. To generate *HCM1* expression plasmids, the *HCM1* open reading frame and endogenous promoter were amplified from the genome and cloned into pRS316. Plasmids for β-galactosidase transcriptional activation assays were generated using the pBTM116 backbone. The mutant or WT *HCM1* TAD region (amino acids 306-511) was amplified by PCR from genomic DNA. The C-terminal 3V5 tag was PCR amplified from the pFA6a-3V5-KanMX6 vector. Both products were purified and cloned pBTM116 using the NEBuilder HiFi DNA assembly kit (New England Biolabs). All plasmids were verified by sequencing. Phosphomutant libraries used for Phosphosite Scanning screens were cloned into a pRS316 vector containing the endogenous *HCM1* promoter upstream of the *HCM1* open reading frame (WT or *hcm1-3N*) in which the region to be mutated had been deleted and replaced by an SphI restriction enzyme site (pRS316-HCM1*SphI or pRS316-hcm1-3N*SphI), to facilitate HiFi DNA assembly. All strains and plasmids generated during this study are available from the corresponding author upon reasonable request.

### Pairwise co-culture competition assays

For pairwise co-culture assays, *GAL1p-HCM1* strains expressing WT GFP or non-fluorescent GFP-Y66F were transformed with pRS316 plasmids expressing the indicated *HCM1* alleles and cultured in synthetic media lacking uracil with 2% galactose. Logarithmic phase cells were mixed in equal proportions by diluting 1 optical density (OD600) of each strain into 10 mL media with galactose. In total, 0.15 optical densities of the mixed culture were then collected, pelleted by centrifugation, resuspended in 2 mL sodium citrate buffer (50 mM sodium citrate, 0.02% $NaN_3$, pH 7.4) and stored at 4 °C, for analysis by flow cytometry at the conclusion of the experiment. Mixed cultures were diluted to 0.006 optical densities in synthetic media lacking uracil with 2% dextrose and allowed to grow for 12 h, during which time the cultures did not reach saturation. Cultures were diluted in this manner every 12 h for a total of 72 h. At each timepoint, 0.15 optical densities were collected and resuspended in 2 mL sodium citrate buffer and stored at 4 °C for subsequent analysis. At the conclusion of the time course, the percentage of each strain in each sample were determined by quantifying the percentage of GFP-positive cells with a Guava EasyCyte HT flow cytometer and GuavaSoft software. In total, 5000 cells were measured in each sample. Data were analyzed with FlowJo software. $N = 3$ technical replicates were performed in each time course and percentages at each timepoint were averaged together. The data presented is an average of $n = 3$ biological replicates.

## Western blotting

Cell pellets (one optical density) were collected for western blotting. Cells were lysed by incubation with 300 μL cold TCA buffer (10 mM Tris pH 8.0, 10% trichloroacetic acid, 25 mM ammonium acetate, 1 mM EDTA) for 10 min on ice. Lysates were then vortexed and pelleted by centrifugation at 16,000×g for 10 min at 4 °C. The supernatant was removed by aspiration and pellets were resuspended in 75 μL resuspension solution (100 mM Tris pH 11, 3% SDS). Samples were then incubated at 95 °C for 5 min, then allowed to cool to room temperature. Lysates were then clarified by centrifugation at 16,000×g for 30 sec. The supernatant was collected and added to 25 μL 4× SDS-PAGE sample buffer (250 mM Tris pH 6.8, 8% SDS, 40% glycerol, 20% β-mercaptoethanol) and heated to 95 °C for 5 min. Samples were then stored at −80 °C.

Standard resolving gels contain 10% acrylamide/bis solution 37.5:1, 0.375 M Tris pH 8.8, 0.1% SDS, 0.1% ammonium persulfate (APS), 0.04% tetremethylethylenediamine (TEMED). Phos-tag gels contain 6% or 8% acrylamide/bis solution 29:1, 386 mM Tris pH 8.8, 0.1% SDS, 0.2% APS, 25 μM Phos-tag acrylamide (Wako), 50 μM manganese chloride and 0.17% TEMED. All stacking gels contain 5% acrylamide/bis solution 37.5:1, 126 mM Tris pH 6.8, 0.1% SDS, 0.1% APS and 0.1% TEMED. Gels were run in 1× running buffer (200 mM glycine, 25 mM Tris, 35 mM SDS) at 140 V (standard) or 150 V (Phos-tag) for two hours. Prior to transferring to nitrocellulose, Phos-tag gels were washed twice with 100 mL 1× transfer buffer with 10 mM EDTA for 15 min, and once with 1× transfer buffer for 10 min. Protein was transferred to nitrocellulose in cold 1× transfer buffer (150 mM glycine, 20 mM Tris, 1.25 mM SDS, 20% methanol) at 0.45 A for 2 h. Nitrocellulose was blocked in 4% milk for 30 min. Proteins were visualized using antibodies that recognize PSTAIRE (P7962, Lot# 015M4840V, Sigma, diluted 1:10,000), V5 (R960-25, Lot# 1923773, 2024280, Invitrogen, diluted 1:1000), and HA (12CA5, gift from David Toczyski, purified from hybridoma, diluted 1:1000). The specificity of antibodies against V5 and HA epitope tags were validated by comparing lysates from yeast strains with each tag to a yeast strain lacking the tags as a negative control. Note that molecular weights are not shown alongside Phos-tag gels since they do not accurately reflect the molecular weights of the proteins.

## Phosphosite scanning library construction

Phosphomutant plasmids were generated by annealing and extending overlapping mutagenic oligonucleotides spanning a 270 base pair region that encompasses all phosphosites, as detailed in Supplementary Fig. 2a. Six oligonucleotides were designed that cover one to three phosphosites each, depending on the distance between phosphosites, with 21–27 base pairs of overlap between consecutive oligonucleotides. For each of the oligonucleotides, several variations were designed to include WT, alanine and glutamic acid substitutions at each phosphosite, in all possible combinations. See Supplementary Table 3 for oligonucleotide sequences. Equimolar ratios of the desired mutagenic oligonucleotides were then combined to a final concentration to 100 μM. For example, to generate the AE library, oligonucleotides encoding A and EE substitutions at each site were combined. 6 μM of pooled oligonucleotides were then phosphorylated at the 5′ ends by incubation with 1 mM ATP, 10 units T4 polynucleotide kinase (New England Biolabs), 1× T4 Polynucleotide Kinase Buffer (New England Biolabs) in a total volume of 50 μL at 37 °C for one hour. Next, oligonucleotides were annealed by diluting 10 μL phosphorylated oligonucleotides to a final volume of 20 μL with sterile water, heating to 95 °C and allowing to slowly cool to 25 °C over the course of an hour. Single-stranded gaps between hybridized oligonucleotides were then filled in by combining 5 μL of the hybridized oligonucleotides from the previous annealing step with 1 unit Q5 High Fidelity DNA polymerase (New England Biolabs), 1× Q5 Reaction Buffer (New England Biolabs), 0.5 mM dNTPs (New England Biolabs), and sterile water to a final volume of 10 μL and incubating at 72 °C for 2 min. Gaps were then ligated by adding 20 units Taq DNA Ligase (New England Biolabs), Taq DNA Ligase Buffer (New England Biolabs), and sterile water to the 10 μL reaction from the previous step to a total volume of 15 μL. The reaction was incubated at 45 °C for 20 min. Finally, the DNA template was amplified by PCR using Phusion High-Fidelity DNA polymerase and Phusion GC Buffer (New England Biolabs). The PCR product was purified using the DNA Clean & Concentrator-25 kit (Zymo Research) and recombined into and SphI digested pRS316-HCM1*SphI using the NEBuilder HiFi DNA assembly kit (New England Biolabs). Plasmids were then transformed into NEB 10-beta Competent E. coli (High Efficiency) (New England Biolabs). All plasmid libraries were constructed in triplicate and mixed prior to transformation to maximize coverage of phosphomutants and avoid bias in oligo mixture or annealing steps.

## Phosphosite scanning screens

Plasmid libraries that were constructed in triplicate were mixed in equal proportion for transformation into cells. Plasmid libraries were transformed into a GAL1p-HCM1 strain grown in YM-1 containing 2% galactose, to maintain the expression of endogenous HCM1. For each screen, wild-type or Hcm1-3N expressing plasmids were added to each library as a control. Transformed cells were grown overnight in 5 mL C-Ura with 2% galactose at 23 °C. After approximately 16 h an aliquot of cells was removed and plated on C-Ura to confirm sufficient transformation efficiency (10X library coverage). The remaining cells were pelleted and washed five times with 15 mL C-Ura with 2% galactose to remove any untransformed plasmid and resuspended in 50 mL C-Ura with 2% galactose and allowed to grow to log phase (~48 h) at 30 °C. Cells were then diluted to an optical density of 0.008 in 75 mL C-Ura with 2% galactose and grown overnight at 30 °C. At the start of the experiment, the population was sampled to determine the representation of each plasmid in the population before selection. Twenty optical densities were collected, frozen on dry ice and stored at −80 °C for subsequent recovery of plasmids and preparation of sequencing libraries. One optical density of cells was harvested for western blotting to confirm levels of plasmid and endogenous Hcm1. Cells were then diluted to an optical density of 0.008–0.016 into C-Ura with 2% dextrose to repress expression of endogenous HCM1 and allowed to grow for 12 h at 30 °C to mid-logarithmic phase. The population was sampled as above and diluted into C-Ura with 2% dextrose every 12 h for a total of 72 h.

## Illumina sequencing library preparation

Plasmids were recovered from frozen cell pellets using the YeaStar Genomic DNA Kit (Zymo Research). Plasmid hcm1 sequence was amplified by PCR for 21 cycles with vector-specific primers using Phusion High-Fidelity DNA polymerase (New England Biolabs). Products were extracted from a 1% agarose gel using the QIAquick Gel Extraction Kit (Qiagen). TruSeq sequencing adapters were added to mutant fragments by PCR using HCM1 specific primers fused to the TruSeq universal adapter or unique TruSeq indexed adapters. Sequences of oligonucleotides used for library construction are included in Supplementary Table 4. Products were extracted from a 1% agarose gel using the QIAquick Gel Extraction Kit (Qiagen). Libraries were pooled, and paired-end 150 base pair sequencing reads obtained by sequencing on a HiSeq4000 platform (Novogene). Due to high sequence identity between samples, all libraries were mixed with at least 50% DNA of heterogeneous sequence to increase diversity.

## Phosphosite scanning data analysis

HCM1 alleles were counted for each paired-end (PE) sequencing fragment that had an exact sequence match in both reads, using a custom python script. Count tables for all screens are included in Supplementary Data 1. Average data from replicate screens are presented in heatmaps (Fig. 2c and Supplementary Figs. 2c, d, 5b, c, and 6), which show the log2 fraction of reads at the indicated timepoints. Primary

data plotted in heatmaps are included in Supplementary Data 2. Selection coefficients were then determined for each mutant by calculating the slope of the log2 fraction of reads versus time for each mutant and subtracting the slope of the log2 fraction of reads versus time for WT (or *hcm1-3N* for the 3N A/E screen). Selection coefficients for all screens are included in Supplementary Data 3. Median selection coefficients for select groups of mutants (with specific numbers of phosphomimetic substitutions, or with specific substitutions at each position) were then calculated and plotted in box and whisker plots using *boxplot()* in R. In all cases, values from all three replicates of each screen are included in the medians. Bar graphs comparing selection coefficients of select mutants show average values from replicate screens.

### Doubling time assays

Cells were grown to mid-logarithmic phase, then diluted to 0.1 optical densities in YM-1 with or without 5 µg/mL nocodazole. Cells were transferred to 96 well plates in triplicate and grown at 30 °C with shaking in a Tecan Infinite M Nano plate reader. Optical densities at 600 nm were measured every 20 min for a total of 33 h. Doubling times were quantified by fitting data points between 0.2 and 0.5 optical densities to an exponential growth equation using GraphPad Prism software. Multiple isolates were tested for each genotype listed.

### β-galactosidase transcriptional activation assays

Cells with an integrated *lacZ* reporter and carrying plasmids expressing LexA-DBD-Hcm1 fusion proteins were grown in synthetic media lacking tryptophan. For each strain, the optical density at 600 nm was measured and 1 mL of cells were collected. Cells were pelleted, and the media was removed. Cells were then resuspended in 0.5 mL Z-buffer (60 mM Na$_2$HPO$_4$ heptahydrate, 40 mM NaH$_2$PO$_4$ monohydrate, 10 mM KCl, 1 mM MgSO$_4$ heptahydrate) and permeabilized through the addition 50 µL chloroform and 10 µL 0.4% SDS followed by vortexing. 300 µL Z-buffer containing 2.4 mg/mL *o*-nitrophenyl-β-d-galactopyranoside and 6 µL/mL β-mercaptoethanol was then added and the time recorded. Reactions were mixed, moved to 30 °C and monitored for a colorimetric change for a maximum of 120 min. Reactions were stopped either when a yellow color change was observed or after 120 min. To stop reactions, 0.5 mL 1 M Na$_2$CO$_3$ was added with vortexing and the time recorded. Completed reactions were stored on ice until all were finished. Chloroform was removed by centrifugation and the optical density of 1 mL of aqueous supernatant was measured at 405 nm. β-galactosidase units were calculated as (1000*OD$_{405}$)/(OD$_{600}$*1 mL*reaction time [min]). All reactions were carried out in triplicate and averaged together. Average values from biological replicates were then averaged together and plotted where indicated.

### Alpha-factor arrest release

Cells were grown to mid-log phase and arrested in G1 phase with 10 µg/mL alpha-factor and released into media without alpha-factor to resume cell cycle progression. Every 15 min, samples were collected for Western blotting and flow cytometry to confirm cell cycle position. Alpha-factor was added back to the synchronized population 45 min after release to prevent cells from entering a second cycle. Cells (0.15 optical densities) were collected at the indicated timepoints, fixed in 70% ethanol, and stored at 4 °C for flow cytometry. Fixed cells were sonicated, treated with 0.25 mg/mL RNase A at 50 °C for an hour, then subsequently treated with 0.125 mg/mL Proteinase K at 50 °C for one hour. Cells were then labeled with 1 µM Sytox Green (Invitrogen). Flow cytometry data were collected using a Millipore Guava easyCyte 5HT with GuavaSoft (v3.3) software. In total, 5000 cells were measured at each timepoint. Data were analyzed using FlowJo software (v10.8.1).

### Reporting summary

Further information on research design is available in the Nature Portfolio Reporting Summary linked to this article.

## Data availability

All sequencing data in this study have been deposited in the NCBI Sequence Read Archive under BioProject # PRJNA841829. Source data are provided with this paper.

## Code availability

Custom scripts used to generate count tables for all screens are publicly available on GitHub at https://github.com/radio1988/mutcount and Zenodo at https://doi.org/10.5281/zenodo.7492458.

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

## Acknowledgements

The authors thank Peter Pryciak, Matthew Winters, Fritz Roth, Julia Flynn, Daniel Bolon, Sneha Gopalan and Tong Wu for reagents and technical advice. We also thank members of the Benanti lab for helpful discussions. This work was supported by National Institutes of Health grants R01GM117152 and R35GM136280 to J.A.B. and R01HD072122 to T.G.F.

## Author contributions

M.M.C. and J.A.B. designed the study. M.M.C. and M.A.N.R. performed the experiments. R.L. generated the count tables from raw sequencing data. L.J.Z. supervised R.L. M.M.C., J.A.B., and T.G.F performed all additional analyses. M.M.C. and J.A.B. wrote the manuscript with input from all authors.

## Competing interests

The authors declare no competing interests.
