## [Peer Review File · Nature Communications]

Phosphosite Scanning reveals a complex phosphorylation code underlying CDK-dependent activation of Hcm1REVIEWER COMMENTS

Reviewer #1 (Remarks to the Author):

Despite our longstanding knowledge that Cdks are the master regulators of the cell cycle, it is still not known how they orchestrate the progression through the cell cycle. The complexity of Cdk regulation is increased by the multisite nature of most of its substrates as the Cdk signal is often processed through a number of phosphorylation sites/steps to result in a multi-phosphorylated output state of a target protein.

In this exciting manuscript, Conti et al. develop a high-throughput Phosphosite Scanning approach to quantitatively address an important question in the field of cell cycle control: Which phosphorylation sites inside a multisite network are physiologically relevant and important for the multisite phosphorylation of a target protein? The developed method allowed authors to directly test the function and importance of each phosphorylation site within a multisite network in cells. As disrupting or mimicking the phosphorylation of Hcm1 transcriptional activation domain leads to opposing effects on protein function and cellular fitness. After screening through libraries of Hcm1 phosphorylation site mutants, Conti and colleagues describe combinations of phosphorylation sites, which not only activate, but also tune the activity of Hcm1 transcriptional activation domain. Importantly, the same scanning approach is easily adaptable for a wide variety of Cdk targets with multisite phosphorylation networks.

I think this manuscript makes a great contribution to the field by: a) establishing the Phosphosite Scanning as a tool to decode the multisite networks of a target protein in cells and b) mapping the mechanism of Hcm1 activation by Cdk-dependent phosphorylation. Therefore, I do think this manuscript meets the standards and is a strong candidate for Nature Communications after the following improvements.

Major points:

- 1) After looking at the Phos-tag western blot images (Fig. 6d and Fig. 7b) in which the Hcm1 wt protein is in highly phosphorylated state. It is surprising that the hcm1-8E mutant has increased fitness compared to the HCM1 in the co-culture assay. As Fig. 6d and Fig. 7b blots are from single timepoint, what is known about the phosphorylation state of Hcm1 throughout the cell cycle and how is Hcm1 binding to the promoter regions controlled/changed in case of hcm1-8E mutant?
- 2) As mentioned above, authors show the phosphorylation status of the full length Hcm1 protein using Phos-tag gels. It would be interesting to see the same for some of the LexA-Hcm1 TAD fusion proteins (Fig. 3a and Fig. 7c). First, this will strengthen the authors claim that phosphorylation is activating transcription independently of DNA binding and that the role of phosphorylation is to promote recruitment of coactivators through conformational change. Second, it will show if the TAD region of Hcm1 has all the elements (cyclin docking sites for example) required for its multisite phosphorylation compared to the full-length protein.
- 3) In Fig. 6a, authors present data from their WT/E screen, showing how the presence of N-terminal phosphorylation sites is important for the Cks1-dependent priming for C-terminally situated phosphorylation sites from the threonine priming site. If their model is right, then the phosphorylation pattern of those remaining sites should be more distributive in Phos-tag gels?
- 4) In Fig. 6d the authors test the lack of phosphorylation for AAWWAAA mutant. It is hard to follow the phosphorylation of one or two extra sites in the presence of multiple sites in Phos-tag gels because of the limited resolution. It would be better to monitor the phosphorylation of those two sites or lack of it in a more minimal setting. For example, adding AAWWAAA back to the 15A mutant.

Minor points:

- 1) In Fig. 1a authors present the structure prediction for the wild type Hcm1 protein. It seems that the key phosphorylation sites T460 and S471 reside in a region, which has propensity to take more structure? In addition, the authors should present similar predictions for some of the T460 and S471 site mutants (changed to EE or AP) used throughout the study as extended data.
- 2) In addition to the mutated eight phosphorylation sites, the TAD region (Fig. 1a) has three additional N-terminally residing SP sites. Is the effect of those sites to the TAD activation known and why they

were left out from the current study? What happens if those sites are mutated?

3) Fig. 2b schematic can be misleading for the readers as it shows HCM1 being off, in reality it is a replacement system. Replacing the wild type with different phosphorylation mutants.

4) It would be more suitable to use the term Cks1-dependent phosphorylation of T and S sites, instead of processive phosphorylation. To my knowledge authors never measure the processivity of Hcm1 TAD's multisite phosphorylation and it's a bit of a stretch to make those conclusions based on only the single time point analyzed in Phos-tag gels.

5) Page 10 lines 270-272. Blocking phosphorylation of these sites with alanine substitutions (hcm1-3N) stabilizes Hcm1 and lengthens its expression through the cell cycle (Fig. S4A)²⁵. Should be changed to:

Blocking phosphorylation of these sites with alanine substitutions (hcm1-3N) stabilizes Hcm1 (Extended Data Fig. 4A) and lengthens its expression through the cell cycle ²⁵.

Fig. 6b presents very clear data supporting the author's conclusion that the N-terminal sites are priming sites for the two physiologically important sites. Very clear result!

Reviewer #2 (Remarks to the Author):

In this manuscript, the multi-site phospho-regulation of a yeast Hcm1 transcription factor is studied using a comprehensive approach that queried the role of every phosphorylation site. The authors have named this approach "Phosphosite Scanning". It was paired with a co-culture and deep sequencing approach that revealed even subtle differences in cellular fitness due to loss or mimic of a phosphorylation event. It is a thorough and complete body of work. While the major findings are not too surprising for a Cdk1 substrate, there are very few examples in the literature in which this high level of rigor has been applied to study whether multiple phosphorylation events on a protein have the same or different functions. The authors learned that there is progressive phosphorylation of the Hcm1 transcriptional activation domain (TAD) mediated likely by the Cdk1 subunit Cks1, there is a threshold of phosphorylation that triggers functional change, the function of HCM1 TAD phosphorylation is independent of DNA binding and modulates HCM1 transcriptional activity, that particular phosphorylation sites are not requisite but preferred, and that phosphomimetic (EE) substitutions do not always work if binding partners (i.e. Cks1) are involved. The work also revealed that tuning phosphorylation of one part of HCM1 (the TAD) is most relevant if the protein is not stabilized by preventing Cdk1 phosphorylation of the degron. While Cdk1 affects two parts of the protein in apparently independent ways, there must be co-regulation of HCM1 stability and function by Cdk1.

The paper is very well-written, the data are carefully presented, and the data analysis are explained well. The data are convincing and reasonably interpreted. While development and implementation of the Phosphosite Scanning approach is a significant focus of the paper, a limitation of this approach is that it can be feasibly used only in yeast or in an in vitro context. Still, it is a comprehensive method that could be applied in other contexts to glean important information about control of a protein's function by protein phosphorylation. I have only minor comments.

Minor comments

1. Extended data Fig.3 were referred to only in the Figure legend as providing additional information; this is true in other cases but it would be helpful if the description indicated what other information would be found. Wonder if it could be incorporated into main text. In the S3 legend, it would also be useful to describe what "W" refers to as I think that is the first instance in the paper that this is used.

2. Line 834: measures should be measured.

3. Optical density was used throughout the methods section. At first instance, please define that this is OD600.

Reviewer #3 (Remarks to the Author):

Conti et al

Conti et al. investigate how multisite phosphorylation controls the function of the yeast transcription factor Hcm1. The authors introduce a method, Phosphosite Scanning, where they make all possible combinations of a phosphomimetic and phosphonull mutants in the region of interest, the activation domain of Hcm1. A strength of the paper is the combination of mutagenesis libraries and biochemical assays, like the Phos-tag Western Blots. The mutagenesis libraries are generally well described and the way the authors present the libraries one at a time makes for a clear and well organized narrative. The authors do a great job going back and forth between analyses that summarize many mutants and inductive reasoning based on a handful of key mutants.

The authors make two claims:

“We show that Phosphosite Scanning identifies multiple combinations of phosphosites that can regulate protein function and reveals specific phosphorylations that are required for phosphorylation at additional sites within a domain.”

and

“Phosphosite Scanning revealed a complex CDK-regulatory circuit that mediates processive phosphorylation of key activating sites in vivo.”

The first claim is well supported by the mutagenesis libraries. There are multiple phosphomimetic mutations that support WT levels of activity. Interestingly phosphorylation of the central two positions or all 6 flanking positions had similar activity levels.

The second claim is very interesting and much harder to definitively prove. The authors provide compelling evidence in support of this claim from the mutagenesis libraries, especially the two libraries that included WT positions. The careful analysis of The Phos-tag Western Blots further support the findings. The evidence that the central positions are phosphorylated when the upstream positions are mutated to alanine was the best support. Finally, the proposed mechanism of sequential by Cks1 phosphorylation is very interesting. We were surprised that the experiment was not repeated in a mutant background, but the claim is well supported without this experiment.

In general, we are initially skeptical of fitness measurements conducted on plasmids, as they are prone to artifacts when the selective coefficients are small. The major effects in this work appear to be large enough to allay our concern. We are curious if the inferred selection coefficients in the pools are correlated with the individual selection coefficients in Figure 1. We recognized this comparison can be difficult in cases where the mean fitness of the population is very different. On a side note, is not always clear if the selection coefficients presented are relative to WT or relative to the population mean. Finally, the use of the GAL1 promoter to control the endogenous Hcm1 was very nice.

The Figures are generally very clear and intuitive. The main exception is panel F of figures 2,4 and 5, which were very cramped and hard to see the effects. For Figure 6, the difference in text widths between W and A/E makes it difficult to look at adjacent sequences. This figure would be more clear as a horizontal barograph, where the sequences could be horizontal and aligned with a fixed width font. This is not a necessary change, but would help the reader understand these plots.

We think it would be really nice to add a multiple sequence alignment of the activation domain to get a sense of whether any of the phosphosites are conserved in position or instead the overall existence of phosphosites is conserved. It would be very nice to have the amino acid sequence of both regions in either the main text figures or the supplement.

When looking at the references and SGD, we saw that additional positions in the TAD are phosphorylated, between the 3 N-terminal sites and the 8 C-terminal sites. Please clarify why only the

8 sites in the activation domain were included the mutagenesis libraries.

Overall, it is a very nice paper. We found it far more interesting than the typical paper in this journal.

There were many sections of the paper that we liked:

-pg 7, paragraph 1 is great.

-Testing the first library in two genetic backgrounds was very interesting. That said, this part could be shortened.

-The observation that DNA binding is disrupted is intriguing. Especially in light of this paper: Krois, A.S., Dyson, H.J., and Wright, P.E. (2018). Long-range regulation of p53 DNA binding by its intrinsically disordered N-terminal transactivation domain. *Proceedings of the National Academy of Sciences* 115, E11302–E11310.

-The nacodozole experiment is very nice

We have some suggestions to make the paper more clear.

-We found this sentence really confusing: “mutation of all eight phosphoacceptor sites within the transactivation domain (TAD) to two glutamic acid residues.” We did not understand it until we saw Figure 2, which is very clear. We think a sentence justifying the choice to make SP>EE vs SP>EP mutations would strengthen the paper.

-Line 150, reference 19 does not seem appropriate for saturating mutagenesis. Perhaps instead: Diss, G., and Lehner, B. (2018). The genetic landscape of a physical interaction. *eLife* 7, 594.

-Line 236 has awkward wording

-This paper might be relevant to cite in the context of multisite phosphorylation conferring more specificity. Lu, Y., Wang, W., and Kirschner, M.W. (2015). Specificity of the anaphase-promoting complex: A single-molecule study. *Science* 348.

-In Figure 6, consider adding WWWEEWWW to the set of mutations shown. Might not be necessary but we were wondering about this one.

REVIEWER COMMENTS

Reviewer #1 (Remarks to the Author):

Despite our longstanding knowledge that Cdks are the master regulators of the cell cycle, it is still not known how they orchestrate the progression through the cell cycle. The complexity of Cdk regulation is increased by the multisite nature of most of its substrates as the Cdk signal is often processed through a number of phosphorylation sites/steps to result in a multi-phosphorylated output state of a target protein.

In this exciting manuscript, Conti et al. develop a high-throughput Phosphosite Scanning approach to quantitatively address an important question in the field of cell cycle control: Which phosphorylation sites inside a multisite network are physiologically relevant and important for the multisite phosphorylation of a target protein? The developed method allowed authors to directly test the function and importance of each phosphorylation site within a multisite network in cells. As disrupting or mimicking the phosphorylation of Hcm1 transcriptional activation domain leads to opposing effects on protein function and cellular fitness. After screening through libraries of Hcm1 phosphorylation site mutants, Conti and colleagues describe combinations of phosphorylation sites, which not only activate, but also tune the activity of Hcm1 transcriptional activation domain. Importantly, the same scanning approach is easily adaptable for a wide variety of Cdk targets with multisite phosphorylation networks.

I think this manuscript makes a great contribution to the field by: a) establishing the Phosphosite Scanning as a tool to decode the multisite networks of a target protein in cells and b) mapping the mechanism of Hcm1 activation by Cdk-dependent phosphorylation. Therefore, I do think this manuscript meets the standards and is a strong candidate for Nature Communications after the following improvements.

We thank the reviewer for supporting publication of our work. We have made all the suggested improvements, as detailed below:

Major points:

1) After looking at the Phos-tag western blot images (Fig. 6d and Fig. 7b) in which the Hcm1 wt protein is in highly phosphorylated state. It is surprising that the hcm1-8E mutant has increased fitness compared to the HCM1 in the co-culture assay. As Fig. 6d and Fig. 7b blots are from single timepoint, what is known about the phosphorylation state of Hcm1 throughout the cell cycle and how is Hcm1 binding to the promoter regions controlled/changed in case of hcm1-8E mutant?

To address this point, we now include a Phos-tag Western blot of WT Hcm1 protein over the cell cycle (new Supplementary Fig. 1). Hcm1 becomes highly phosphorylated as cells enter S-phase (15-30 minutes after G1 release) and is degraded as cells enter mitosis (45 min after release). This is consistent with *in vitro* data that shows that Hcm1 is efficiently phosphorylated by Cln2/CDK, which is active in early S phase (Kõivomägi et al., *NSMB*, 2013).

We apologize for not including information regarding hcm1-8E DNA binding in the first version of the MS. We previously found that hcm1-8E binds at higher levels to target gene promoters *in vivo* (Landry et al, *EMBO J*, 2014; this mutant is referred to as hcm1-16E in that paper because the 8 phosphosites were mutated to 16 E residues to mimic the charge of phosphorylation). This

suggests that the WT protein is not phosphorylated on all 8 sites in the TAD at any one time. We now include additional information about the 8E mutant on p5.

2) As mentioned above, authors show the phosphorylation status of the full length Hcm1 protein using Phos-tag gels. It would be interesting to see the same for some of the LexA-Hcm1 TAD fusion proteins (Fig. 3a and Fig. 7c). First, this will strengthen the authors claim that phosphorylation is activating transcription independently of DNA binding and that the role of phosphorylation is to promote recruitment of coactivators through conformational change. Second, it will show if the TAD region of Hcm1 has all the elements (cyclin docking sites for example) required for its multisite phosphorylation compared to the full-length protein.

We thank the reviewer for the excellent suggestion and have added Phos-tag blots of several LexA-Hcm1 TAD fusions in new Supplementary Fig. 4d. We specifically examined mutants that include combinations of WT sites and A substitutions, since these are most informative. Consistent with our model, mutants that cannot activate transcription (hcm1-8A and AAAWWAAA) are not phosphorylated, whereas mutants that are active in transactivation assays do display phosphoshifts (WWWAWWW, WWWAWWWW, WWWWAWWW, as well as WT). This result confirms that this TAD fragment contains all the sequence elements required for phosphorylation of the domain.

3) In Fig. 6a, authors present data from their WT/E screen, showing how the presence of N-terminal phosphorylation sites is important for the Cks1-dependent priming for C-terminally situated phosphorylation sites from the threonine priming site. If their model is right, then the phosphorylation pattern of those remaining sites should be more distributive in Phos-tag gels?

Our expectation is that if N-terminal sites are mutated to E, we would observe either more distributive phosphorylation, as the reviewer suggests, or possibly an absence of C-terminal phosphorylation if a phosphorylated threonine is required for phosphorylation of more C-terminal sites. Unfortunately, we cannot analyze E mutants in Phos-tag gels, since E substitutions cause the protein to migrate aberrantly, which is observed even in normal SDS-PAGE gels. However, our expectations are the same when the N-terminal sites are mutated to A. We have now included a Phos-tag analysis of a mutant that has alanine substitutions at the first three positions and is WT at the remaining sites (AAAWWWW; with the WT sites added back to the 15A mutant, as suggested below). We find that there is very little change in the phosphoshift in this mutant, beyond what is seen in the 15A mutant, although there may be a slight increase in phosphorylation at one or two sites (new Fig 6e, light exposure). These data suggest that the N-terminal sites are required for most phosphorylation at sites located more C-terminally.

4) In Fig. 6d the authors test the lack of phosphorylation for AAAWWAAA mutant. It is hard to follow the phosphorylation of one or two extra sites in the presence of multiple sites in Phos-tag gels because of the limited resolution. It would be better to monitor the phosphorylation of those two sites or lack of it in a more minimal setting. For example, adding AAAWWAAA back to the 15A mutant.

As requested, we have now integrated this mutant into an Hcm1 protein that lacks all other S/T-P sites. As described above, we also constructed the AAAWWWWW mutant in this background. As shown in Fig 6e, both mutants look very similar to the 15A mutant, although there is slightly more phosphorylation of AAAWWWWW compared to AAAWWAAA (described above). In contrast, the hcm1-7N protein (which contains all S/T-P sites in the TAD but no other S/T-Ps) is

highly phosphorylated. As noted in point 2 above, we have also examined the AAAWWAAA mutant in the context of the LexA-DBD fusion proteins and see the same result. These data confirm our conclusion that the three N-terminal sites in the TAD are required for phosphorylation of more C-terminal sites.

Minor points:

1) In Fig. 1a authors present the structure prediction for the wild type Hcm1 protein. It seems that the key phosphorylation sites T460 and S471 reside in a region, which has propensity to take more structure? In addition, the authors should present similar predictions for some of the T460 and S471 site mutants (changed to EE or AP) used throughout the study as extended data.

It is true that the key sites T460 and T471 are in a region that is predicted to be more structured. We discuss this observation on p19. At the reviewer's suggestion we examined disorder predictions for hcm1-8A and hcm1-8E mutants (new Supplementary Fig. 7b). The introduction of these mutations had only very modest effects on the disorder plots and did not change the prediction of the short, ordered region.

2) In addition to the mutated eight phosphorylation sites, the TAD region (Fig. 1a) has three additional N-terminally residing SP sites. Is the effect of those sites to the TAD activation known and why they were left out from the current study? What happens if those sites are mutated?

We apologize for not explaining this better. In our previous study, we found that mutation of just the 8 C-terminal sites was sufficient to confer benomyl sensitivity and downregulation of target gene expression and that this mutant is indistinguishable from an hcm1 delete strain (Landry et al, *EMBO J*, 2014; referred to as hcm1-8C in that paper). From these data we concluded that the additional three sites do not contribute to Hcm1 activation and have not included them in this study.

3) Fig. 2b schematic can be misleading for the readers as it shows HCM1 being off, in reality it is a replacement system. Replacing the wild type with different phosphorylation mutants.

We have edited the figure to clarify that it is the WT Hcm1 protein that is shut off by the carbon shift.

4) It would be more suitable to use the term Cks1-dependent phosphorylation of T and S sites, instead of processive phosphorylation. To my knowledge authors never measure the processivity of Hcm1 TAD's multisite phosphorylation and it's a bit of a stretch to make those conclusions based on only the single time point analyzed in Phos-tag gels.

We have reworded the text throughout to remove claims of processive phosphorylation, as suggested.

5) Page 10 lines 270-272. Blocking phosphorylation of these sites with alanine substitutions (hcm1-3N) stabilizes Hcm1 and lengthens its expression through the cell cycle (Fig. S4A)25. Should be changed to: Blocking phosphorylation of these sites with alanine substitutions (hcm1-3N) stabilizes Hcm1 (Extended Data Fig. 4A) and lengthens its expression through the cell cycle

We have corrected this sentence as suggested.

Fig. 6b presents very clear data supporting the author's conclusion that the N-terminal sites are priming sites for the two physiologically important sites. Very clear result!

We thank the reviewer for their enthusiasm, thorough review, and helpful suggestions.

Reviewer #2 (Remarks to the Author):

In this manuscript, the multi-site phospho-regulation of a yeast Hcm1 transcription factor is studied using a comprehensive approach that queried the role of every phosphorylation site. The authors have named this approach "Phosphosite Scanning". It was paired with a co-culture and deep sequencing approach that revealed even subtle differences in cellular fitness due to loss or mimic of a phosphorylation event. It is a thorough and complete body of work. While the major findings are not too surprising for a Cdk1 substrate, there are very few examples in the literature in which this high level of rigor has been applied to study whether multiple phosphorylation events on a protein have the same or different functions. The authors learned that there is progressive phosphorylation of the Hcm1 transcriptional activation domain (TAD) mediated likely by the Cdk1 subunit Cks1, there is a threshold of phosphorylation that triggers functional change, the function of HCM1 TAD phosphorylation is independent of DNA binding and modulates HCM1 transcriptional activity, that particular phosphorylation sites are not requisite but preferred, and that phosphomimetic (EE) substitutions do not always work if binding partners (i.e. Cks1) are involved. The work also revealed that tuning phosphorylation of one part of HCM1 (the TAD) is most relevant if the protein is not stabilized by preventing Cdk1 phosphorylation of the degron. While Cdk1 affects two parts of the protein in apparently independent ways, there must be co-regulation of HCM1 stability and function by Cdk1.

The paper is very well-written, the data are carefully presented, and the data analysis are explained well. The data are convincing and reasonably interpreted. While development and implementation of the Phosphosite Scanning approach is a significant focus of the paper, a limitation of this approach is that it can be feasibly used only in yeast or in an in vitro context. Still, it is a comprehensive method that could be applied in other contexts to glean important information about control of a protein's function by protein phosphorylation. I have only minor comments.

We thank the reviewer for supporting publication of our work. All the suggested edits have been made (addressed below).

Minor comments

1. Extended data Fig.3 were referred to only in the Figure legend as providing additional information; this is true in other cases but it would be helpful if the description indicated what other information would be found. Wonder if it could be incorporated into main text. In the S3 legend, it would also be useful to describe what "W" refers to as I think that is the first instance in the paper that this is used.

We have made the suggested edits and now include specific information about what information can be found in Supplementary Files when each is cited.

2. Line 834: measures should be measured.

This error has been corrected.

3. Optical density was used throughout the methods section. At first instance, please define that this is OD600.

This is now defined at the first use of "optical density".

Reviewer #3 (Remarks to the Author):

Conti et al

Conti et al. investigate how multisite phosphorylation controls the function of the yeast transcription factor Hcm1. The authors introduce a method, Phosphosite Scanning, where they make all possible combinations of a phosphomimetic and phosphonull mutants in the region of interest, the activation domain of Hcm1. A strength of the paper is the combination of mutagenesis libraries and biochemical assays, like the Phos-tag Western Blots. The mutagenesis libraries are generally well described and the way the authors present the libraries one at a time makes for a clear and well organized narrative. The authors do a great job going back and forth between analyses that summarize many mutants and inductive reasoning based on a handful of key mutants.

The authors make two claims:

“We show that Phosphosite Scanning identifies multiple combinations of phosphosites that can regulate protein function and reveals specific phosphorylations that are required for phosphorylation at additional sites within a domain.”

and

“Phosphosite Scanning revealed a complex CDK-regulatory circuit that mediates processive phosphorylation of key activating sites in vivo.”

The first claim is well supported by the mutagenesis libraries. There are multiple phosphomimetic mutations that support WT levels of activity. Interestingly phosphorylation of the central two positions or all 6 flanking positions had similar activity levels.

The second claim is very interesting and much harder to definitively prove. The authors provide compelling evidence in support of this claim from the mutagenesis libraries, especially the two libraries that included WT positions. The careful analysis of The Phos-tag Western Blots further support the findings. The evidence that the central positions are phosphorylated when the upstream positions are mutated to alanine was the best support. Finally, the proposed mechanism of sequential by Cks1 phosphorylation is very interesting. We were surprised that the experiment was not repeated in a mutant background, but the claim is well supported without this experiment.

We agree it would be a nice addition to include data on Hcm1 phosphorylation in a *cks1* mutant strain and have attempted experiments in temperature-sensitive mutants. However, even at the permissive temperature Cks1 function is compromised in these strains and the cells grow extremely poorly, which makes it exceedingly difficult to detect Hcm1 expression. For this reason, we have not been able to examine Hcm1 phosphorylation in *cks1* mutants.

In general, we are initially skeptical of fitness measurements conducted on plasmids, as they are prone to artifacts when the selective coefficients are small. The major effects in this work appear to be large enough to allay our concern. We are curious if the inferred selection coefficients in the pools are correlated with the individual selection coefficients in Figure 1. We recognized this comparison can be difficult in cases where the mean fitness of the population is very different. On a side note, it is not always clear if the selection coefficients presented are relative to WT or relative to the population mean. Finally, the use of the GAL1 promoter to control the endogenous Hcm1 was very nice.

We have now included a comparison of selection coefficients calculated from pairwise and pooled assays and have included this in new Supplementary Fig. 2e. Both assays result in highly similar values, although the pooled assays may skew slightly more positive.

We apologize for not making it clear how selection coefficients are normalized. With one exception, all selection coefficients are calculated relative to WT. The only exception is for the 3N screen presented in Fig. 4, where SCs were calculated relative to the 3N allele (which is stabilized, but all sites in the TAD are WT). In this instance we could not normalize to WT because the fully WT allele was not included in the screen. We have edited the text to make sure these points are clear.

The Figures are generally very clear and intuitive. The main exception is panel F of figures 2, 4 and 5, which were very cramped and hard to see the effects. For Figure 6, the difference in text widths between W and A/E makes it difficult to look at adjacent sequences. This figure would be more clear as a horizontal barograph, where the sequences could be horizontal and aligned with a fixed width font. This is not a necessary change, but would help the reader understand these plots.

We have changed all the indicated plots to a horizontal orientation, as requested.

We think it would be really nice to add a multiple sequence alignment of the activation domain to get a sense of whether any of the phosphosites are conserved in position or instead the overall existence of phosphosites is conserved. It would be very nice to have the amino acid sequence of both regions in either the main text figures or the supplement.

We thank the reviewer for the suggestion and now include a multiple sequence alignment of the Hcm1 TAD from budding yeasts in new Supplementary Fig. 7a. This alignment highlights the fact that the positions and identities of the first 5 S/T-P sites in the TAD are highly conserved, whereas the 3 C-terminal sites are less so. This is consistent with our conclusion that the first three sites are essential for Cks1 docking and the fourth and fifth sites contribute the greatest amount to Hcm1 activation.

When looking at the references and SGD, we saw that additional positions in the TAD are phosphorylated, between the 3 N-terminal sites and the 8 C-terminal sites. Please clarify why only the 8 sites in the activation domain were included in the mutagenesis libraries.

As mentioned above, we previously showed that only the 8 most C-terminal sites are required for Hcm1 activation, and that mutation of these sites alone inactivates the protein. We now clarify this point in the text on p4.

Overall, it is a very nice paper. We found it far more interesting than the typical paper in this journal.

There were many sections of the paper that we liked:

-pg 7, paragraph 1 is great.

-Testing the first library in two genetic backgrounds was very interesting. That said, this part could be shortened.

-The observation that DNA binding is disrupted is intriguing. Especially in light of this paper: Krois, A.S., Dyson, H.J., and Wright, P.E. (2018). Long-range regulation of p53 DNA binding by its intrinsically disordered N-terminal transactivation domain. *Proceedings of the National Academy of Sciences* 115, E11302–E11310.

-The nacodozole experiment is very nice

We have some suggestions to make the paper more clear.

-We found this sentence really confusing: “mutation of all eight phosphoacceptor sites within the transactivation domain (TAD) to two glutamic acid residues.” We did not understand it until we saw Figure 2, which is very clear. We think a sentence justifying the choice to make SP>EE vs SP>EP mutations would strengthen the paper.

We have edited and expanded this section of the text for clarity, as suggested.

-Line 150, reference 19 does not seem appropriate for saturating mutagenesis. Perhaps instead: Diss, G., and Lehner, B. (2018). The genetic landscape of a physical interaction. *eLife* 7, 594.

We have added the suggested citation.

-Line 236 has awkward wording

We have edited this sentence to be clearer.

-This paper might be relevant to cite in the context of multisite phosphorylation conferring more specificity. Lu, Y., Wang, W., and Kirschner, M.W. (2015). Specificity of the anaphase-promoting complex: A single-molecule study. *Science* 348.

We agree that there are interesting parallels between multisite phosphorylation and multisite ubiquitination. We decided against including this citation since it would require additional discussion and the text is already quite long.

-In Figure 6, consider adding WWWEWWWW to the set of mutations shown. Might not be necessary but we were wondering about this one.

We have added this mutant to Fig 6a, as requested.

REVIEWERS' COMMENTS

Reviewer #1 (Remarks to the Author):

I thank the authors for their discussion and additional experiments to address my initial comments. I do not have any further comments/questions and I support publication of their work in Nature Communications.

Reviewer #3 (Remarks to the Author):

The initial manuscript was a strong piece of work. I found the revised manuscript improved and easier to read. I am grateful for many of changes made by the authors to present the data more clearly. In particular, I found the supplemental figures greatly enhanced and easier to follow.

I fully support acceptance of the revised manuscript for publication.

Sections that were improved:

-Lines 160-162 were very clear.

-Figure S1 is great

-Figure S2 is fabulous

-Figure S3 is lovely.

-I really liked that the 5 N-terminal phosphosites, which are more important in the mutagenesis, showed higher conservation in Figure S7a.

-The additional phospho-gels really improved the paper.

I have one very one minor comment: pg. 5 line 136-145. I found this section to be a much more clear description of what the mutations are. However, the first time I read it, I was a little confused if hcm1-8E referred to the 8xEP or the 8xEE mutants. The second time I read the sentence I figured it out. It is fine as is, but might more clear if the (hcm1-8E) was defined in line 17, after the E-E.

Response to Reviewers

Reviewer #1 (Remarks to the Author):

I thank the authors for their discussion and additional experiments to address my initial comments. I do not have any further comments/questions and I support publication of their work in Nature Communications.

We thank the reviewer for supporting publication of our work.

Reviewer #3 (Remarks to the Author):

The initial manuscript was a strong piece of work. I found the revised manuscript improved and easier to read. I am grateful for many of changes made by the authors to present the data more clearly. In particular, I found the supplemental figures greatly enhanced and easier to follow.

I fully support acceptance of the revised manuscript for publication.

Sections that were improved:

-Lines 160-162 were very clear.

-Figure S1 is great

-Figure S2 is fabulous

-Figure S3 is lovely.

-I really liked that the 5 N-terminal phosphosites, which are more important in the mutagenesis, showed higher conservation in Figure S7a.

-The additional phospho-gels really improved the paper.

We thank the reviewer for supporting publication of our work.

I have one very one minor comment: pg. 5 line 136-145. I found this section to be a much more clear description of what the mutations are. However, the first time I read it, I was a little confused if hcm1-8E referred to the 8xEP or the 8xEE mutants. The second time I read the sentence I figured it out. It is fine as is, but might more clear if the (hcm1-8E) was defined in line 17, after the E-E.

We have edited this sentence for clarity.